# Spectral Compressive Imaging via Chromaticity-Intensity Decomposition

**Xiaodong Wang** [1,2], **Zijun He** [1,2], **Ping Wang** [2],
**Lishun Wang** [3], **Yanan Hu** [1,2], **Xin Yuan** [2,*]
[1] Zhejiang University,
[2] School of Engineering, Westlake University,
[3] Chengdu Institute of Biology, Chinese Academy of Sciences

## Abstract

In coded aperture snapshot spectral imaging (CASSI), the captured measurement entangles spatial and spectral information, posing a severely ill-posed inverse problem for hyperspectral images (HSIs) reconstruction. Moreover, the captured radiance inherently depends on scene illumination, making it difficult to recover the intrinsic spectral reflectance that remains invariant to lighting conditions. To address these challenges, we propose a chromaticity-intensity decomposition framework, which disentangles an HSI into a spatially smooth intensity map and a spectrally variant chromaticity cube. The chromaticity encodes lighting-invariant reflectance, enriched with high-frequency spatial details and local spectral sparsity. Building on this decomposition, we develop CIDNet—a Chromaticity-Intensity Decomposition unfolding network within a dual-camera CASSI system. CIDNet integrates a hybrid spatial-spectral Transformer tailored to reconstruct fine-grained and sparse spectral chromaticity and a degradation-aware, spatially-adaptive noise estimation module that captures anisotropic noise across iterative stages. Extensive experiments on both synthetic and real-world CASSI datasets demonstrate that our method achieves superior performance in both spectral and chromaticity fidelity. Code is released at: `https://github.com/xiaodongwo/CIDNet`.

## 1 Introduction

Coded aperture snapshot spectral imaging (CASSI) has emerged as a promising architecture for capturing hyperspectral images (HSIs) in a single shot [1, 25, 34]. By jointly modulating the spectral cube with a coded aperture and dispersing it spatially through a prism, CASSI produces a 2D compressed measurement that encodes both spatial and spectral information. This compressive measurement fuses (shears) the spectral bands, making each pixel of the 2D sensor a mixture of many wavelengths. As a result, recovering the full 3D spectral image becomes a severely under-determined, ill-posed inverse problem.

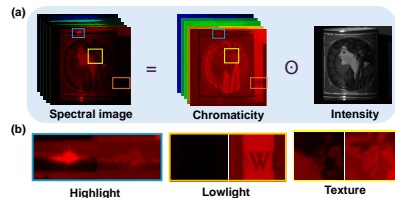

Figure 1: (a) Chrmaticity-Intensity decomposition of HSI images (b) Chromaticity exhibits highlight removal, lowlight enhancement and high-frequency textures.

The difficulty comes mainly from two aspects. One is that **spatial and spectral signals are highly overlapped and entangled in the compressed measurement.** Many works attempt to address this through various priors or deep models, broadly categorized into four paradigms. Optimization-based methods [17, 33] introduce hand-crafted priors such as total variation or low-rank constraints.

---

*Corresponding Author: Xin Yuan (xyuan@westlake.edu.cn)

39th Conference on Neural Information Processing Systems (NeurIPS 2025).

However, their performance is often limited in recovering spatial structures, especially under complex textures or noise. Plug-and-Play (PnP) approaches [24, 39] integrate powerful pre-trained denoisers into iterative solvers, yet these methods typically denoise each or few spectral band independently, neglecting spectral correlation and structure. Deep unfolding methods [4, 7, 14, 19, 37, 38] bridge model-based and data-driven paradigms by learning iterative modules guided by the CASSI physics. End-to-end networks [3, 10, 20, 21, 26] leverage CNN or Transformer to directly infer the spectral cube from measurements. These deep learning-based frameworks implicitly exploit spatial-spectral dependencies and have shown promising performance. Recently, diffusion model [22, 30, 35] and Mamba[23] have been used for spectral reconstruction. Nonetheless, all these approaches rely on network backbones to learn spatial-spectral features in an implicit manner. There lacks a clear and interpretable decomposition or quantitative structure that explicitly characterizes the physical roles of spatial and spectral components during reconstruction.

The second challenge is that **existing methods often overlook the impact of illumination.** Since the captured spectral measurement is radiance-based, it inherently entangles the intrinsic surface reflectance with scene illumination. This coupling makes the reconstruction sensitive to lighting variations across time and environments, thereby limiting spectral accuracy. To address similar issues in the RGB image, prior works have explored intrinsic image decomposition [2, 8, 15, 16] and Retinex-based models [28, 29] to explicitly separate reflectance from illumination, enabling applications such as shadow removal and low-light enhancement. In the hyperspectral remote sensing community, several studies have also extended intrinsic decomposition to spectral reflectance and illumination separation [11, 12, 32], offering better invariance to lighting conditions. However, to the best of our knowledge, such decomposition has not yet been incorporated into CASSI reconstruction.

In this paper, we propose a novel chromaticity learning framework for compressive spectral imaging, which leverages a chromaticity-intensity decomposition prior under the CASSI sensing mechanism. Our motivation is illustrated in Fig. 1, where the spectral image cube $\mathbf{X}$ is factorized as:

$$\mathbf{X} = \mathbf{C} \odot \mathbf{I}, \tag{1}$$

where $\mathbf{C}$ denotes the *chromaticity cube* and $\mathbf{I}$ represents the *intensity image*. Notably, $\mathbf{C}$ exhibits several desirable properties: (i) spatially invariant to illumination, suppressing highlights and enhancing details in low-light regions; (ii) spectrally sparse with localized support (as illustrated latter); (iii) enriched with high-frequency texture, essential for fine-detail recovery. In contrast, the intensity component $\mathbf{I}$ captures the global illumination structure in the scene. It is interesting to note that the chromaticity exhibits more intrinsic characteristics of the sample compared to hyperspectral images. Hence, learning the chromaticity instead of HSIs seems to benefit the field more.

Building upon the above observations, we propose a physically interpretable chromaticity-intensity decomposition model tailored for CASSI systems. By leveraging a dual-camera CASSI setup, we validate this decomposition paradigm within both traditional optimization-based solvers and deep unfolding frameworks. To further explore its potential, we design a novel Chromaticity-Intensity Decomposition Network (CIDNet), which incorporates the spectral sparsity of chromaticity through a sparse `TopK` spectral Transformer, and models spatially anisotropic noise via a degradation-aware, spatially-adaptive variance estimator.

In summary, our main contributions are summarized as follows:

i) We propose a novel **chromaticity-intensity decomposition model** for spectral compressive imaging, which explicitly separates hyperspectral images into lighting-invariant chromaticity and smooth intensity. We further validate its effectiveness on optimization-based and unfolding algorithms in a dual-camera CASSI setting.

ii) We develop an **intensity-guided deep unfolding network** that incorporates the chromaticity decomposition into unfolding algorithm. The network features a hybrid spatial-spectral Transformer (HSST) architecture, where the encoder leverages window-based local spatial attention (Spa-LWSA) and the decoder employs sparse `TopK` spectral attention (Spec-TKSA) to capture localized spectral structures.

iii) We introduce a degradation-aware, spatially-adaptive **dual noise estimation module** (DNEM) to model anisotropic noise across different reconstruction stages. This module enables each iteration to adaptively handle varying noise levels across spatial locations.

iv) Extensive experiments on both synthetic and real datasets demonstrate that our method achieves state-of-the-art performance in terms of spectral reconstruction and chromaticity fidelity.

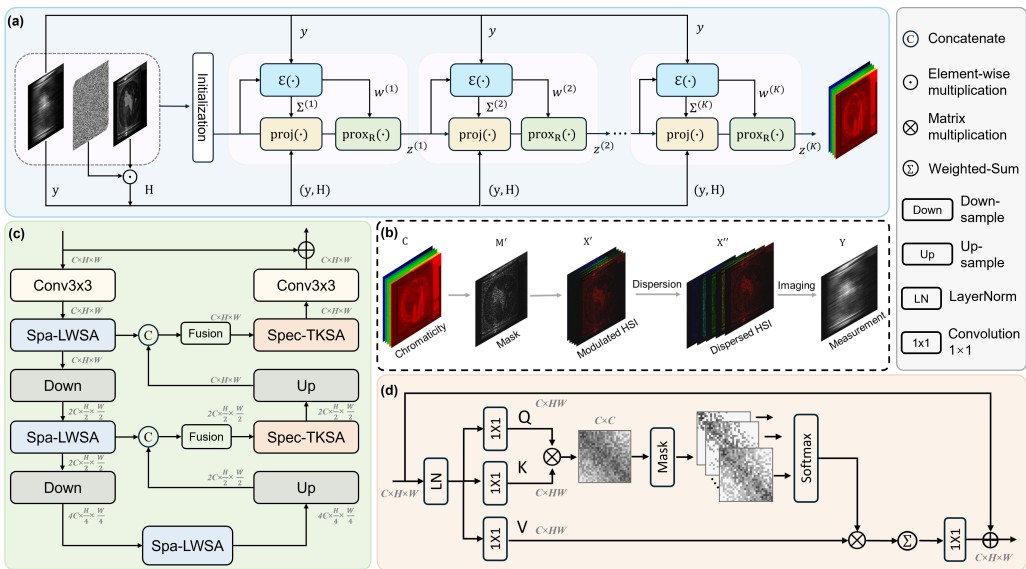

Figure 2: (a) The architecture of our CIDNet with $K$ stages (iterations). (b) The CASSI system uses an intensity-guided mask to modulate the chromaticity. (c) Diagram of asymmetric backbone for our hybrid spatial-spectral Transformer (HSST), with a local window spatial attention (Spa-LWSA) in Encoder and sparse TopK spectral attention module (Spec-TKSA) in Decoder. (d) Details of Spec-TKSA.

## 2 Proposed Method

### 2.1 Degradation Model of CASSI

Inspired by chromaticity-intensity decomposition in RGB intrinsic image analysis, we extend this concept to the hyperspectral domain. Given a hyperspectral image cube $\mathbf{X} \in \mathbb{R}^{H \times W \times N_\lambda}$, we decompose it into a spatially smooth intensity image $\mathbf{I} \in \mathbb{R}^{H \times W}$ and a chromaticity cube $\mathbf{C} \in \mathbb{R}^{H \times W \times N_\lambda}$ as:

$$\mathbf{X}(u, v, \lambda) = \mathbf{C}(u, v, \lambda) \odot \mathbf{I}(u, v), \tag{2}$$

where $(u, v)$ denotes spatial location, $\lambda$ is the spectral band index, and $\odot$ denotes pixel-wise multiplication. Specifically, the intensity image is defined as the average spectral energy per pixel:

$$\mathbf{I}(u, v) = \frac{1}{N_\lambda} \sum_{\lambda=1}^{N_\lambda} \mathbf{X}(u, v, \lambda), \tag{3}$$

We found that intensity image can be approximated as a PAN image in dual-camera CASSI. A detailed proof in this PAN-Intensity Equivalence is provided in supplement materials. Hence, the chromaticity is computed as the normalized spectral signature:

$$\mathbf{C}(u, v, \lambda) = \frac{\mathbf{X}(u,v,\lambda)}{\mathbf{I}(u,v)+\epsilon}, \tag{4}$$

where $\epsilon$ is a small constant to avoid division by zero. This decomposition separates the multiplicative effect of illumination $\mathbf{I}(u, v)$ from the spectral reflectance $\mathbf{C}(u, v, \lambda)$, which captures intrinsic scene properties. Importantly, $\mathbf{C}$ is invariant to changes in illumination intensity and direction, enabling more robust modeling of reflectance and spectral reconstruction under varying lighting conditions (see supplement materials). After decomposition, the CASSI measurement process can be modeled as follows. The hyperspectral cube $\mathbf{X}$ is modulated by a coded aperture $\mathbf{M} \in \mathbb{R}^{H \times W}$, resulting in a spatially coded cube:

$$\mathbf{X}'(u, v, \lambda) = \mathbf{C}(u, v, \lambda) \odot \mathbf{I}(u, v) \odot \mathbf{M}(u, v). \tag{5}$$

Now we can treat $\mathbf{I}(u, v) \odot \mathbf{M}(u, v)$ as a new formation of coded mask $\mathbf{M}'(u, v) = \mathbf{I}(u, v) \odot \mathbf{M}(u, v)$ incorporating the spatial intensity, we call it intensity-guided mask. This leaves the chromaticity an unknown variable when the intensity is obtained beforehand. Following a typical CASSI formulation, the modulated cube $\mathbf{X}'$ is then passed through a dispersive element that shifts each spectral band $\lambda_{n_\lambda}$ by a wavelength-dependent displacement $d(\lambda_{n_\lambda} - \lambda_c)$ along the spatial axis (e.g., the $x$-axis). The sheared datacube can be expressed as:

$$\mathbf{X}''(u, v, n_\lambda) = \mathbf{X}'(u, v + d(\lambda_{n_\lambda} - \lambda_c), \lambda_{n_\lambda}), \tag{6}$$

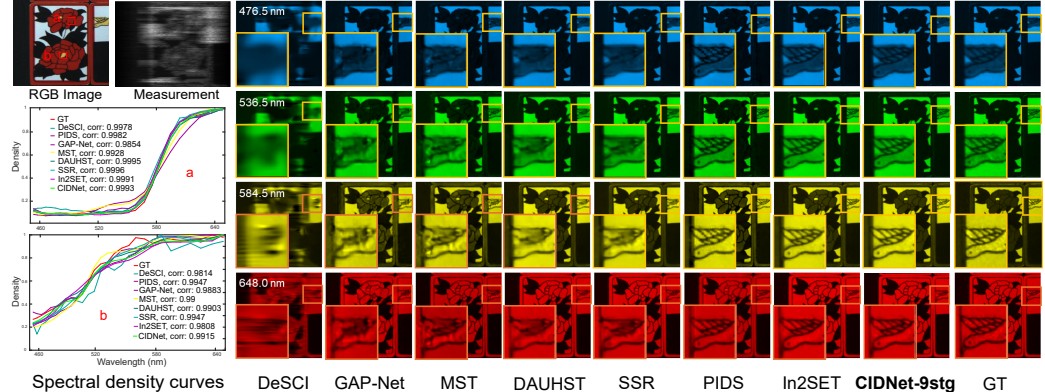

Figure 3: Simulation HSIs reconstruction comparisons of Scene 7 with 4 (out of 28) spectral channels. The left shows the spectral curves corresponding to the two red boxes of the RGB image. The top-right depicts the enlarged patches corresponding to the yellow boxes in the bottom HSIs. Zoom in for a better view.

where $\lambda_c$ is the reference wavelength that remains unshifted. Finally, the 2D measurement $\mathbf{Y} \in \mathbb{R}^{H \times (W + d \cdot (N_\lambda - 1))}$ acquired by the camera is a summation of all dispersed bands:

$$\mathbf{Y}(u, v) = \sum_{n_\lambda = 1}^{N_\lambda} \mathbf{X}''(u, v, n_\lambda) + \mathbf{N}(u, v), \tag{7}$$

where $\mathbf{N}$ is additive measurement noise. This can be written compactly in vectorized form as:

$$\mathbf{y} = \mathbf{\Phi}(\mathbf{c} \odot \mathbf{i}) + \mathbf{n}, \tag{8}$$

where $\mathbf{c} = \mathrm{vec}(\mathbf{C})$, $\mathbf{i} = \mathrm{vec}(\mathbf{I})$, $\mathbf{\Phi}$ is the sensing matrix determined by the modulation and dispersion process, and $\mathbf{n}$ is the vectorized noise term. Given that the intensity map $\mathbf{I}$ is known (PAN or RGB image in dual-camera CASSI scheme), we formulate the chromaticity-based measurement model as a standard linear inverse problem:

$$\mathbf{y} = \mathbf{Hc} + \mathbf{n}, \tag{9}$$

where $\mathbf{c}$ denotes the vectorized chromaticity, $\mathbf{H}$ is the effective sensing matrix that incorporates both the CASSI modulation-dispersion process and the known intensity modulation, and $\mathbf{n}$ is the vectorized noise.

## 2.2  Optimization Framework of CASSI

To characterize realistic imaging noise, we assume an anisotropic Gaussian noise model $\mathbf{n} \sim \mathcal{N}(\mathbf{0}, \mathbf{\Sigma})$, where $\mathbf{\Sigma} = \mathrm{diag}(\sigma_1^2, \ldots, \sigma_n^2)$ is a diagonal covariance matrix whose diagonal entries $\sigma_i^2$ represent the noise variance at the $i$-th pixel. This implies that the noise is spatially varying but uncorrelated across pixels. Under a Bayesian framework, the posterior probability

Table 1: Data-consistency projection comparison.

| Method | Gradient projection updating |
|---|---|
| ISTA [36] | $\mathbf{c}^{k+1} = \mathbf{z}^k + \mathbf{H}^\top (\mathbf{y} - \mathbf{Hz}^k)$ |
| GAP [19] | $\mathbf{c}^{k+1} = \mathbf{z}^k + \mathbf{H}^\top (\mathbf{HH}^\top)^{-1}(\mathbf{y} - \mathbf{Hz}^k)$ |
| HQS [4] | $\mathbf{c}^{k+1} = \mathbf{z}^k + \mathbf{H}^\top (\mathbf{HH}^\top + \mu \mathbb{I})^{-1}(\mathbf{y} - \mathbf{Hz}^k)$ |
| **Ours** | $\mathbf{c}^{k+1} = \mathbf{z}^k + \mathbf{H}^\top (\mathbf{HH}^\top + \mu \mathbf{\Sigma})^{-1}(\mathbf{y} - \mathbf{Hz}^k)$ |

of the chromaticity $\mathbf{c}$ is given by $p(\mathbf{c} \mid \mathbf{y}) \propto p(\mathbf{y} \mid \mathbf{c}) \cdot p(\mathbf{c})$, and the likelihood is:

$$p(\mathbf{y} \mid \mathbf{c}, \mathbf{\Sigma}) \propto \exp\left(-\tfrac{1}{2}(\mathbf{y} - \mathbf{Hc})^\top \mathbf{\Sigma}^{-1}(\mathbf{y} - \mathbf{Hc})\right), \tag{10}$$

and $p(\mathbf{c}) \propto \exp(-\tau R(\mathbf{c}))$ is a generic prior over chromaticity with regularization function $R(\cdot)$ and weight $\tau > 0$. Maximizing the posterior leads to the following Maximum A Posteriori (MAP) estimation problem:

$$\hat{\mathbf{c}} = \mathrm{argmin}_{\mathbf{c}} \ \tfrac{1}{2}(\mathbf{y} - \mathbf{Hc})^\top \mathbf{\Sigma}^{-1}(\mathbf{y} - \mathbf{Hc}) + \tau R(\mathbf{c}). \tag{11}$$

When the noise is homoscedastic (i.e., $\sigma_i = 1$), the problem reduces to the common quadratic form: $\hat{\mathbf{c}} = \mathrm{argmin}_{\mathbf{c}} \ \tfrac{1}{2}\|\mathbf{y} - \mathbf{Hc}\|_2^2 + \tau R(\mathbf{c})$. We rewrite Eq. (11) using an auxiliary variable $\mathbf{z}$:

$$\hat{\mathbf{c}}, \hat{\mathbf{z}} = \mathrm{argmin}_{\mathbf{c},\mathbf{z}} \ \tfrac{1}{2}(\mathbf{y} - \mathbf{Hc})^\top \mathbf{\Sigma}^{-1}(\mathbf{y} - \mathbf{Hc}) + \tau R(\mathbf{z}), \quad \text{s.t.} \quad \mathbf{c} = \mathbf{z}. \tag{12}$$

Using the half-quadratic splitting (HQS) framework, Eq. (12) is minimized by solving the following data-consistency and data-prior subproblems iteratively:

$$\mathbf{c}^{(k+1)} = \mathrm{argmin}_{\mathbf{c}} \ \tfrac{1}{2}(\mathbf{y} - \mathbf{Hc})^\top \mathbf{\Sigma}^{-1}(\mathbf{y} - \mathbf{Hc}) + \tfrac{\mu}{2}\|\mathbf{c} - \mathbf{z}^{(k)}\|_2^2, \tag{13}$$

$$\mathbf{z}^{(k+1)} = \mathrm{argmin}_{\mathbf{z}} \ \tfrac{\mu}{2}\|\mathbf{z} - \mathbf{c}^{(k+1)}\|_2^2 + \tau R(\mathbf{z}), \tag{14}$$

where $\mu$ is a penalty parameter and $k$ represent $k$th iteration. The **c**-subproblem in Eq. (13) is quadratic and has a closed-form solution:

$$\mathbf{c}^{(k+1)} = \left(\mathbf{H}^\top \boldsymbol{\Sigma}^{-1} \mathbf{H} + \mu \mathbb{I}\right)^{-1} \left(\mathbf{H}^\top \boldsymbol{\Sigma}^{-1} \mathbf{y} + \mu \mathbf{z}^{(k)}\right), \tag{15}$$

where $\mathbb{I}$ represent identity matrix. Note that $\mathbf{H}^\top \boldsymbol{\Sigma}^{-1} \mathbf{H}$ is a fat matrix and $(\mathbf{H}^\top \boldsymbol{\Sigma}^{-1} \mathbf{H} + \mu \mathbb{I})^{-1}$ will be difficult to compute and thus we simplify it based on the Sherman-Morrison-Woodbury formula,

$$\left(\mathbf{H}^\top \boldsymbol{\Sigma}^{-1} \mathbf{H} + \mu \mathbb{I}\right)^{-1} = \mu^{-1}\mathbb{I} - \mu^{-2}\mathbf{H}^\top \left(\boldsymbol{\Sigma} + \mu^{-1}\mathbf{H}\mathbf{H}^\top\right)^{-1} \mathbf{H}. \tag{16}$$

In CASSI systems, $\mathbf{H}\mathbf{H}^\top$ is a diagonal matrix defined as $\mathbf{H}\mathbf{H}^\top \triangleq \mathrm{diag}\{h_1, \ldots, h_n\}$. With $\boldsymbol{\Sigma} \triangleq \mathrm{diag}(\sigma_1^2, \ldots, \sigma_n^2)$ and detailed derivation in supplement materials, we obtain a generalized form of gradient projection, which is expressed as,

$$\mathbf{c}^{(k+1)} = \mathbf{z}^{(k)} + \mathbf{H}^\top (\mathbf{H}\mathbf{H}^\top + \mu\boldsymbol{\Sigma})^{-1}(\mathbf{y} - \mathbf{H}\mathbf{z}^{(k)}). \tag{17}$$

Define this gradient projection as $\mathbf{c}^{(k+1)} = \mathrm{proj}_{\boldsymbol{\Sigma}}(\cdot)$. Interestingly, we observed that this gradient projection resembles previous optimization-based methods but introduces a key change relates to the spatially-varying noise modeling. As summarized in Tab. 1, while traditional ISTA [36], GAP [19], and HQS [4] methods employ static data-consistency steps with fixed regularization, our method proposes a dynamic, spatially-adaptive correction mechanism. Finally, we update $\mathbf{z}^{(k+1)}$ using any proximal operator depending on the prior $R(\cdot)$,

$$\mathbf{z}^{(k+1)} = \mathrm{prox}_{\tau/\mu \cdot R}(\mathbf{c}^{(k+1)}). \tag{18}$$

If the noise variance $\boldsymbol{\Sigma}$ is known a priori, with Eq. (17) and Eq. (18), this concludes the efficient HQS derivation with anisotropic Gaussian noise. However, in practice, the noise map is unavailable and may vary dynamically across iterations (e.g., in PnP or unfolding methods). To effectively account for the degradation-varying characteristics in the CASSI system, we parameterize the anisotropic noise covariance $\boldsymbol{\Sigma}^{(k)}$ and the denoising strength $\tau_k/\mu_k$ in a stage-specific manner. Both parameters are learned by a degradation-aware estimator $\mathcal{E}$ that takes as input the current iterate $\mathbf{z}^{(k)}$ and the measurement $\mathbf{y}$:

$$\{\boldsymbol{\Sigma}^{(k)}, \tau_k/\mu_k\} = \mathcal{E}(\mathbf{z}^{(k)}, \mathbf{y}). \tag{19}$$

The estimator $\mathcal{E}$ is implemented as a lightweight CNN that jointly captures spatial structure in $\mathbf{z}^{(k)}$ and the encoded degradation in $\mathbf{y}$. A detailed network module is found in supplement materials. We denote $\boldsymbol{\Sigma}^{(k)}$ and $\omega^{(k)} = \tau^{(k)}/\mu^{(k)}$ as the noise map for gradient projection and proximal mapping (denoiser) respectively (Dual Noise-Estimation Module (DNEM), as we refered to). The estimated $\boldsymbol{\Sigma}^{(k)}$ reflects the anisotropic uncertainty in the current iterate, and modulates the linear update of $\mathbf{c}^{(k+1)}$ via Eq. (17), while $\omega^{(k)}$ controls the noise level fed into the proximal denoiser for $\mathbf{z}^{(k+1)}$ via Eq. (18). The final iterative process can be expressed as:

$$\begin{aligned}
\{\boldsymbol{\Sigma}^{(k)}, \omega^{(k)}\} &= \mathcal{E}(\mathbf{z}^{(k)}, \mathbf{y}), \\
\mathbf{c}^{(k+1)} &= \mathrm{proj}_{\boldsymbol{\Sigma}^{(k)}}(\mathbf{z}^{(k)}) = \mathbf{z}^{(k)} + \mathbf{H}^\top(\mathbf{H}\mathbf{H}^\top + \boldsymbol{\Sigma}^{(k)})^{-1}(\mathbf{y} - \mathbf{H}\mathbf{z}^{(k)}), \\
\mathbf{z}^{(k+1)} &= \mathrm{prox}_{\omega^{(k)} \cdot R}(\mathbf{c}^{(k+1)}).
\end{aligned} \tag{20}$$

Here, $\mu^{(k)}$ is omitted due to the usage of the network, $R(\cdot)$ is a regularization prior, which could be total variation, or a learned denoiser as in our experiments. This stage-adaptive formulation enables flexible and efficient recovery under spatially variant degradation patterns.

To validate the effectiveness of our chromaticity-intensity decomposition strategy, we explore two integration paradigms: a traditional model-based iterative scheme and a deep unfolding network. The classical iterative algorithm is described in supplement materials, leveraging analytical priors and explicit update rules based on the degradation model. In contrast, our primary design adopts a learnable unfolding structure, as illustrated in Fig. 2. Each stage is composed of a learnable noise estimation, an analytical reconstruction step shown in Eq. (17) and a learned proximal denoiser. This framework offers the interpretability of traditional optimization while benefiting from the expressiveness and efficiency of deep networks.

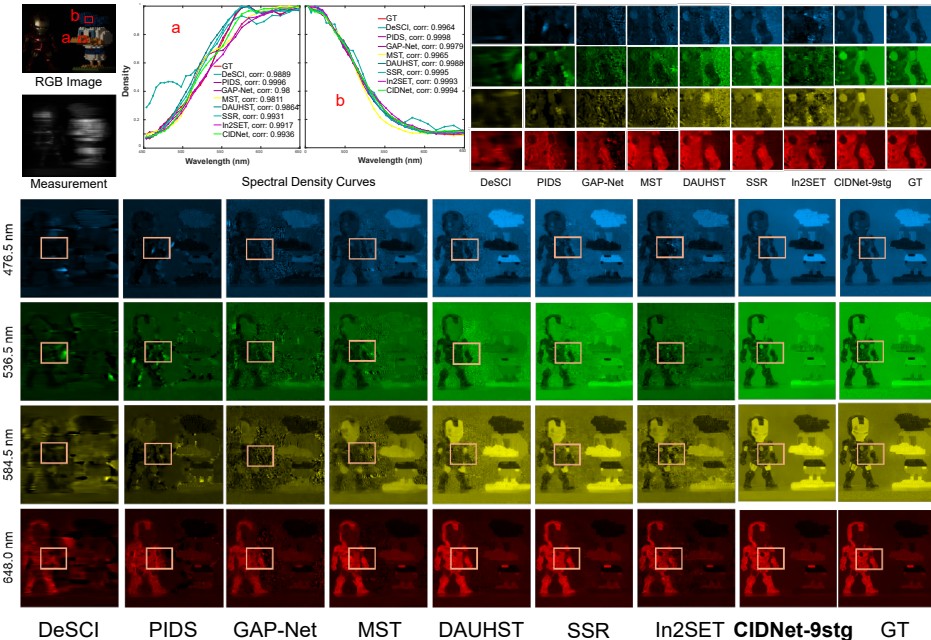

Figure 4: Simulation: **chromaticity** reconstruction of Scene 8 with 4 (out of 28) spectral channels. The spectral curves correspond to the two red boxes in the RGB image (top-middle). The top-right depicts the zoomed patches corresponding to the yellow boxes in the bottom chromaticity.

## 2.3 Hybrid Spatial-Spectral Transformer

To better reconstruct the chromaticity component $\mathbf{C}$, which inherently contains rich spatial textures and locally correlated spectral patterns (illustrated in Fig. 5), we propose an asymmetric UNet backbone using Hybrid Spatial-Spectral Transformer (HSST). This module is specifically designed to simultaneously learn the high-frequency details in spatial dimensions and sparse-local dependencies in the spectral domain, as shown in Fig. 2.

We adopt a dual-branch design: the spatial attention branch captures intra-image textures through a Swin Transformer in Encoder, while the spectral attention branch is tailored to exploit sparse and locally correlated spectral features using a TopK spectral attention mechanism in Decoder. This asymmetric design is inspired by [37] and motivated by experimental verification, where the asymmetric design has better reconstruction results.

**Spectral Attention.** Unlike the spectral correlation in HSIs, chromaticity spectra features exhibit structured sparsity and localized correlation, see Fig. 5. Motivated by this, we introduce a window-based spectral TopK attention mechanism, where attention is applied across the spectral channels within each local spatial window. Specifically, each spectral token attends only to its $K$ most relevant spectral neighbors, enforcing both sparsity and locality.

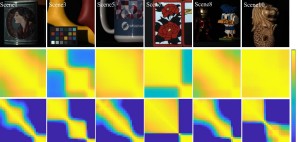

Figure 5: Demonstration of sparse and local spectral correlation of chromaticity. Top: RGB contents of the benchmark testing data. Middle: spectral correlation coefficient matrices of the HSIs (28×28). Bottom: Corresponding matrices by the chromaticity.

Given an input feature cube $\mathbf{X} \in \mathbb{R}^{H \times W \times C}$, we first divide it into non-overlapping spatial windows of size $N \times N$, resulting in a batch of local cubes $\{\mathbf{X}_w\} \subset \mathbb{R}^{N^2 \times C}$. Within each window, we perform spectral self-attention across the $C$ channels for every spatial location. We begin by computing the query, key, and value embeddings using learned $1 \times 1$ convolutions:

$$\{\mathbf{Q_i}, \mathbf{K_i}, \mathbf{V_i}\} = \text{Conv}_{1 \times 1}(\mathbf{X}_w) \in \mathbb{R}^{N^2 \times C \times d}, \tag{21}$$

where $d$ is the embedding dimension per head. To model inter-channel dependencies, we transpose the last two dimensions and perform attention along the channel axis. For each position $i \in \{1, ..., N^2\}$, the attention is computed as:

$$\mathbf{A}_i = \text{Softmax} \left( \text{TopK}(\frac{\mathbf{Q}_i \mathbf{K}_i^\top}{\sqrt{d}}) \right) \in \mathbb{R}^{C \times C}, \tag{22}$$

Table 2: Comparisons of **HSIs** between CIDNet and SOTA methods on KAIST simulation dataset. PSNR (upper entry in each cell), and SSIM (lower entry in each cell) are reported. The best result is highlighted in bold.

| Method | Params(M) | GFlOPs(G) | S1 | S2 | S3 | S4 | S5 | S6 | S7 | S8 | S9 | S10 | Avg |
|---|---|---|---|---|---|---|---|---|---|---|---|---|---|
| DeSCI [17] | - | - | 27.13 | 23.04 | 26.62 | 34.96 | 23.94 | 22.38 | 24.45 | 22.03 | 24.56 | 23.59 | 25.27 |
| | | | 0.748 | 0.620 | 0.818 | 0.897 | 0.706 | 0.0.683 | 0.743 | 0.673 | 0.732 | 0.587 | 0.721 |
| GAP-Net [19] | 4.27 | 78.58 | 33.74 | 33.26 | 34.28 | 41.03 | 31.44 | 32.40 | 32.27 | 30.46 | 33.51 | 30.24 | 33.26 |
| | | | 0.911 | 0.900 | 0.929 | 0.967 | 0.919 | 0.925 | 0.902 | 0.905 | 0.915 | 0.895 | 0.917 |
| MST-L [3] | 2.03 | 28.15 | 35.40 | 35.87 | 36.51 | 42.27 | 32.77 | 34.80 | 33.66 | 32.67 | 35.39 | 32.50 | 35.18 |
| | | | 0.941 | 0.944 | 0.953 | 0.973 | 0.947 | 0.955 | 0.925 | 0.948 | 0.949 | 0.941 | 0.948 |
| DAUHST-9stg [4] | 6.15 | 79.50 | 37.25 | 39.02 | 41.05 | 46.15 | 35.80 | 37.08 | 37.57 | 35.10 | 40.02 | 34.59 | 38.36 |
| | | | 0.958 | 0.967 | 0.971 | 0.983 | 0.969 | 0.970 | 0.963 | 0.966 | 0.970 | 0.956 | 0.967 |
| SSR-9stg [38] | 5.18 | 78.93 | 39.07 | 42.04 | 44.49 | 48.80 | 38.64 | 38.50 | 39.16 | 36.96 | 43.12 | 36.08 | 40.69 |
| | | | 0.970 | 0.981 | 0.980 | 0.990 | 0.980 | 0.978 | 0.971 | 0.976 | 0.980 | 0.968 | 0.978 |
| PIDS-RGB [5] | - | - | 42.09 | 40.08 | 41.50 | 48.55 | 40.05 | 39.00 | 36.63 | 37.02 | 38.82 | 38.64 | 40.24 |
| | | | 0.983 | 0.949 | 0.968 | 0.989 | 0.982 | 0.974 | 0.940 | 0.948 | 0.953 | 0.980 | 0.967 |
| In2SET-9stg [27] | 9.69 | 59.40 | 42.56 | 46.42 | 44.55 | **50.63** | 42.01 | 42.49 | 41.59 | 40.53 | 43.83 | **42.33** | 43.69 |
| | | | 0.989 | 0.994 | **0.986** | **0.996** | 0.992 | 0.991 | 0.983 | 0.989 | 0.990 | **0.994** | 0.990 |
| **CIDNet-3stg** | **1.40** | **24.80** | 40.88 | 45.39 | 43.55 | 47.54 | 40.37 | 41.94 | 40.98 | 41.11 | 42.52 | 40.79 | 42.51 |
| | | | 0.986 | 0.993 | 0.983 | 0.993 | 0.990 | 0.901 | 0.981 | 0.992 | 0.987 | 0.992 | 0.989 |
| **CIDNet-5stg** | 2.33 | 41.26 | 41.56 | 46.36 | 43.98 | 47.92 | 41.47 | 42.27 | 41.27 | 41.36 | 43.90 | 40.56 | 43.07 |
| | | | 0.987 | 0.994 | 0.984 | 0.993 | 0.992 | 0.992 | 0.982 | 0.992 | 0.990 | 0.992 | 0.990 |
| **CIDNet-7stg** | 3.26 | 57.71 | 41.66 | 46.79 | 44.52 | 48.51 | 41.44 | 42.56 | 41.46 | 41.93 | 44.36 | 41.49 | 43.47 |
| | | | 0.988 | 0.995 | 0.985 | 0.994 | 0.992 | 0.993 | 0.983 | 0.993 | 0.990 | 0.993 | 0.990 |
| **CIDNet-9stg** | 4.19 | 74.16 | **42.72** | **47.88** | **44.87** | 48.83 | **42.59** | **43.01** | **42.28** | **42.26** | **44.68** | 42.05 | **44.12** |
| | | | **0.990** | **0.996** | **0.986** | 0.994 | **0.993** | **0.993** | **0.985** | **0.994** | **0.991** | **0.994** | **0.991** |

where $\texttt{TopK}(\cdot)$ retains only the $\texttt{TopK}$ values per row and masks out the rest with $-\infty$ before applying softmax. This yields a sparse attention map across spectral channels for each spatial location $i$ in the window $\mathbf{Z}_i = \mathbf{A}_i \mathbf{V}_i \in \mathbb{R}^{C \times d}$. To improve the robustness and expressiveness of spectral modeling, we further adopt a multi-ratio strategy. Instead of selecting a single sparsity level, we generate multiple attention maps using different $\texttt{TopK}$ ratios (e.g., $\{1/2, 2/3, 3/4, 4/5\}$ of $C$), and then aggregate them adaptively. Specifically, let $\mathbf{A}^{(r)}$ denote the sparse attention computed under ratio $r$, and $\alpha^{(r)}$ be a learnable scalar weight. The final attention output is then formulated as:

$$\mathbf{Z}_i = \sum_{r \in \mathcal{R}} \alpha^{(r)} \cdot \text{Softmax}(\mathbf{A}_i^{(r)}) \mathbf{V}_i, \tag{23}$$

where $\mathcal{R}$ denotes the set of $\texttt{TopK}$ ratios. This fusion across multiple sparsity levels enables the network to capture both dominant and complementary spectral correlations, further improving spectral detail preservation. and reshaped back to the original window structure. After processing all windows, the outputs are stitched to reconstruct the full feature map. This spectral $\texttt{TopK}$ attention not only reduces the computational burden from dense $O(C^2)$ to $O(KC)$, but also explicitly captures the structured sparsity observed in reflectance spectra—where only a few wavelengths contribute significantly. Empirically, we find that this strategy enhances spectral sharpness and suppresses irrelevant cross-band mixing. Following the standard Transformer architecture, we apply a conventional feedforward network after the sparse $\texttt{TopK}$ attention, which is not the focus of our work.

**Spatial Attention.** Inspired by recent advances in Swin Transformers [18], we employ the Swin Transformer as our spatial modeling backbone. The input feature map $\mathbf{X} \in \mathbb{R}^{H \times W \times C}$ is divided into non-overlapping windows of size $M \times M$. Within each window, we perform multi-head self-attention (MSA) by computing

$$\mathbf{Z}_{\text{spa}} = \text{MSA}\left(\text{LN}(\mathbf{X})\right) + \mathbf{X}, \quad \mathbf{Z}_{\text{out}} = \text{FFN}\left(\text{LN}(\mathbf{Z}_{\text{spa}})\right) + \mathbf{Z}_{\text{spa}}, \tag{24}$$

where $\text{LN}(\cdot)$ denotes layer normalization, and $\texttt{FFN}$ is a standard feedforward network. The relative position bias and shifted-window mechanism in Swin Transformer enhance local texture modeling while maintaining global continuity across windows.

## 3 Experiments

We conduct comprehensive experiments on both simulated and real-world CASSI systems. The datasets, training settings, and implementation details are introduced as follows.

Table 3: Comparisons of **intensity (left)** and **chromaticity (right)** between CIDNet and SOTA methods on KAIST simulation dataset. PSNR (upper entry in each cell), and SSIM (lower entry in each cell) are reported.

| Method | S1 | S2 | S3 | S4 | S5 | S6 | S7 | S8 | S9 | S10 | Avg |
|---|---|---|---|---|---|---|---|---|---|---|---|
| DeSCI [17] | 29.42/14.76 0.84/0.49 | 27.48/23.02 0.72/0.67 | 31.01/26.75 0.92/0.83 | 41.71/19.31 0.97/0.75 | 26.62/20.14 0.81/0.72 | 25.12/17.66 0.79/0.50 | 27.08/20.87 0.82/0.60 | 24.55/19.48 0.77/0.54 | 29.19/24.90 0.86/0.79 | 25.61/10.95 0.69/0.32 | 28.78/19.79 0.82/0.62 |
| GAP-Net [19] | 36.78/21.85 0.95/0.62 | 35.84/20.88 0.93/0.61 | 39.31/25.66 0.98/0.79 | 45.90/18.63 0.98/0.66 | 33.94/23.30 0.94/0.76 | 34.37/19.43 0.95/0.57 | 35.65/24.43 0.95/0.64 | 32.57/20.31 0.94/0.54 | 37.69/17.34 0.95/0.52 | 32.05/13.19 0.92/0.39 | 36.41/20.50 0.95/0.61 |
| MST-L [3] | 38.05/23.95 0.96/0.72 | 38.42/30.26 0.96/0.80 | 41.07/32.36 0.98/0.89 | 47.73/19.59 0.99/0.75 | 35.14/28.89 0.96/0.84 | 36.44/29.68 0.97/0.78 | 37.04/27.18 0.96/0.75 | 34.89/25.94 0.96/0.75 | 39.26/20.65 0.97/0.74 | 34.42/17.56 0.95/0.60 | 38.25/25.61 0.97/0.76 |
| DAUHST-9stg [4] | 39.68/25.72 0.97/0.77 | 40.84/29.54 0.97/0.85 | 45.08/34.98 0.99/0.94 | 49.24/31.54 0.99/0.86 | 37.84/34.08 0.98/0.91 | 38.79/32.63 0.98/0.84 | 40.33/31.95 0.98/0.83 | 36.71/31.39 0.97/0.80 | 44.55/34.30 0.98/0.90 | 35.91/26.88 0.96/0.66 | 40.90/31.30 0.98/0.84 |
| SSR-9stg [38] | 41.19/24.45 0.98/0.83 | 43.35/31.26 0.98/0.93 | 48.50/40.51 0.99/0.97 | 50.01/37.65 0.99/0.95 | 40.85/33.46 0.98/0.95 | 40.13/36.05 0.98/0.92 | 41.86/32.72 0.98/0.86 | 38.44/33.03 0.98/0.91 | 45.98/37.76 0.99/0.95 | 37.09/27.86 0.97/0.86 | 42.74/33.47 0.98/0.91 |
| PIDS [5] | 37.44/18.20 0.99/0.72 | 38.81/17.27 0.98/0.33 | 34.81/22.74 0.98/0.80 | 42.83/17.92 0.98/0.68 | 33.71/19.89 0.98/0.78 | 36.66/15.90 0.98/0.32 | 35.95/19.10 0.97/0.61 | 36.81/17.51 0.98/0.36 | 37.06/16.24 0.98/0.37 | 35.43/9.37 0.98/0.15 | 36.95/17.41 0.98/0.51 |
| In2SET-9stg [27] | 58.06/29.42 1.00/0.77 | 59.61/31.15 1.00/0.83 | 59.60/36.02 1.00/0.93 | 62.63/25.56 1.00/0.81 | 57.55/33.62 1.00/0.90 | 58.55/27.68 1.00/0.79 | 57.77/32.05 1.00/0.83 | 57.82/26.56 1.00/0.76 | 59.44/25.26 1.00/0.84 | 56.52/12.93 1.00/0.49 | 58.75/28.03 1.00/0.80 |
| **CIDNet-3stg** | 63.78/28.02 1.00/0.83 | 64.70/31.10 1.00/0.93 | 63.18/39.77 1.00/0.97 | **67.12**/36.14 1.00/0.96 | 61.77/37.54 1.00/0.96 | 64.73/35.88 1.00/0.93 | 62.72/32.71 1.00/0.85 | 65.24/33.97 1.00/0.94 | **63.65**/37.60 1.00/0.95 | 64.93/33.15 1.00/0.89 | 64.18/34.59 1.00/0.92 |
| **CIDNet-5stg** | **64.67**/29.70 1.00/0.85 | **65.40**/33.41 1.00/0.94 | **63.41**/39.54 1.00/0.97 | 66.83/38.02 1.00/0.96 | **64.03**/38.46 1.00/0.96 | **65.85**/36.36 1.00/0.93 | **63.18**/32.70 1.00/0.86 | **65.84**/36.70 1.00/0.94 | 63.65/38.45 1.00/0.96 | **64.98**/32.59 1.00/0.89 | **64.78**/35.59 1.00/0.93 |
| **CIDNet-7stg** | 61.89/**30.81** 1.00/0.85 | 62.96/32.06 1.00/0.94 | 60.99/41.17 1.00/0.97 | 65.42/38.06 1.00/0.96 | 61.26/36.91 1.00/0.96 | 63.87/36.49 1.00/0.93 | 60.32/33.23 1.00/0.86 | 63.88/35.98 1.00/0.95 | 61.14/**38.80** 1.00/0.96 | 62.52/33.43 1.00/0.89 | 62.43/35.69 1.00/0.93 |
| **CIDNet-9stg** | 62.28/25.89 1.00/0.86 | 63.54/**34.29** 1.00/0.95 | 61.42/**41.48** 1.00/0.97 | 65.86/38.22 1.00/0.96 | 61.26/**39.71** 1.00/0.97 | 64.00/**37.10** 1.00/0.94 | 60.80/**33.31** 1.00/0.87 | 64.13/36.69 1.00/0.95 | 62.02/38.00 1.00/0.96 | 62.71/**33.43** 1.00/0.90 | 62.80/**35.81** 1.00/0.93 |

**Simulation Dataset.** We adopt two widely used hyperspectral datasets: **CAVE** [31] and **KAIST** [6]. The CAVE dataset contains 32 hyperspectral images with a spatial resolution of $512 \times 512$. The KAIST dataset provides 30 high-resolution hyperspectral scenes of size $2704 \times 3376$. We obtain ground-truth multi-spectral chromaticity and intensity using chromaticity-intensity decomposition Eq. (28). Following prior works [20, 21, 3], we use all CAVE images for training and select 10 scenes from KAIST for evaluation.

**Implementation Details.** Our model is implemented in PyTorch and trained using the Adam optimizer [13] for 300 epochs. The initial learning rate is set to $4 \times 10^{-4}$ and updated using a cosine annealing schedule. We employ the $\ell_2$ loss between the reconstructed chromaticity and ground-truth chromaticity as the objective function. For training, we randomly extract 3D hyperspectral patches from each scene. For simulated data, the patch size is $256 \times 256 \times 28$, for real-world data, we use patches of size $350 \times 260 \times 26$. In simulation experiments, the forward imaging model is configured with a dispersion shift step $d = 2$, directing dispersion along the horizontal axis (rightward). In real-world scenarios, we assume a vertical dispersion direction and set $d = 1$, consistent with the dual-camera hardware setup. All experiments are conducted in Nvidia A40 GPU.

**Comparing Methods.** We compared the HSIs and chromaticity reconstruction performance of our CIDNet with other 7 SOTA methods, including DeSCI[17], GAP-Net[19], MST-L[3], DAUHST-9stg[4], SSR-9stg[38] and two dual-camera CASSI algorithm: PIDS[5] and In2SET-9stg[27]. PIDS is compared with RGB image as guidance (in original paper). We evaluate the reconstruction performance from two perspectives: hyperspectral images (HSIs) and chromaticity. Unlike the baseline methods that directly reconstruct HSIs, our method assumes that the intensity is known and focuses on reconstructing the chromaticity. For a fair comparison in the HSI domain, we

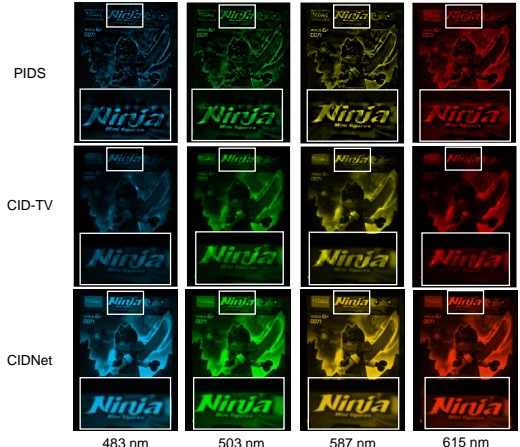

Figure 6: Real-data reconstruction in dual-camera CASSI.

obtain our reconstructed HSIs by multiplying the recovered chromaticity with the known intensity. Conversely, for chromaticity-level comparison, we perform chromaticity–intensity decomposition on the HSIs reconstructed by the baseline methods to extract their chromaticity components. The reconstruction quality of HSIs is evaluated using peak signal-to-noise ratio (PSNR) and structural similarity index (SSIM).

Table 4: Break-down ablation study on individual components of the proposed method.

| Base-1 | Int. | HSST | DNEM | PSNR | SSIM | Params | FLOPs |
|:---:|:---:|:---:|:---:|:---:|:---:|:---:|:---:|
| ✓ | | | | 35.77 | 0.949 | 1.11 | 16.13 |
| ✓ | ✓ | | | 40.83 | 0.984 | 1.11 | 16.13 |
| ✓ | ✓ | ✓ | | 42.30 | 0.988 | 1.44 | 25.04 |
| ✓ | ✓ | ✓ | ✓ | **42.51** | **0.989** | 1.40 | 24.80 |

Table 5: Break-down ablation study on spectral self-attention mechanism.

| Method | Base-2 | WSSA-WSSA | TKSA-TKSA | LWSA+TKSA | LWSA-TKSA |
|:---|:---:|:---:|:---:|:---:|:---:|
| PSNR | 40.95 | 41.98 | 42.05 | 42.36 | **42.51** |
| SSIM | 0.985 | 0.988 | 0.988 | 0.988 | **0.989** |
| Params | 1.12 | 1.33 | 1.33 | 1.27 | 1.40 |
| FLOPs | 18.03 | 23.29 | 25.16 | 23.47 | 24.80 |

Table 6: Break-down ablation study on intensity-guided mask.

| Method | ADMM | ADMM-Int | MST | MST-Int | DAUHST | DAUHST-Int |
|:---|:---:|:---:|:---:|:---:|:---:|:---:|
| PSNR | 24.80 | 37.09 | 34.26 | 37.28 | 37.21 | 42.64 |
| SSIM | 0.712 | 0.965 | 0.935 | 0.971 | 0.959 | 0.989 |

## 3.1 Quantitative Results

As shown in Tab. 2, we compare the PSNR and SSIM of HSIs with SOTA methods. our proposed CIDNet excels in 8 out of 10 scenes, particularly in CIDNet-9stg, achieving an average PSNR of 44.12dB and SSIM of 0.991. This significantly surpasses previous unfolding and end-to-end networks, and also in dual-camera CASSI systems, such as In2SET-9stg. A visual comparison is shown in Fig. 3. We provide the visual comparison of simulation Scene7 with 4 out of 28 spectral channels. In addition, we plot the spectral density curves of two regions in the top left RGB image. Our CIDNet-9stg achieves relatively higher spectral accuracy with reference spectra, demonstrating the effectiveness of our method. To verify the reconstructed quality of chromaticity and intensity, we compare the PSNR and SSIM of reconstructed chromaticity and intensity (with decomposition of HSIs), which is shown in Table 3. Note that our method assumes a known intensity. Therefore, to ensure a fair comparison of intensity, we decompose the reconstructed HSIs to extract their intensity component for evaluation. Our CIDNet achieves significant improvements in metrics of chromaticity and intensity. For dual-camera CASSI algorithm PIDS and In2SET, we achieve the best chromaticity metrics with a PSNR of 35.81 and a SSIM of 0.93. A visual comparison of reconstructed chromaticity is shown in Fig. 4. In this research, we used a real-world DCCHI measurement Ninja, taken from publicly available data as detailed in [9]. Fig. 6 illustrates the reconstruction results for four spectral bands in this scene, using two dual-camera CASSI reconstruction algorithms, PIDS and CID-TV, where CID-TV is the iterative CID algorithm using Total Variationa as Regularizer, details can be found in supplement materials. The comparison highlights the superior image restoration quality of our model over other methods, validating its effectiveness and reliability in real-world applications.

## 3.2 Ablation Study

**Effectiveness of Intensity, HSST and DNEM.** We verify the effectiveness of our proposed intensity-guided mask and two network module, HSST and DNEM. We adopt Base-1, derived by retaining binary mask and removing spatial-spectral attention and noise estimation from CIDNet-3stg to conduct the ablation study, where we used ground-truth HSIs instead of chromaticity for supervision. Tab. 4 shows the results of PSNR and SSIM of different settings, and our method achieves a significant 6.74dB PSNR improvements compared with Base-1.

**Robustness of intensity-guided mask.** We test the robustness of intensity-guided mask $M'$ by employing this intensity mask to ADMM, DAUHST and MST, the result is shown in Tab. 6. We compare the reconstruction quality with/without intensity-guided mask and find that it is significantly improving the base (without intensity) in iterative, end-to-end and unfolding framework. Note that DAUHST achieves slightly higher metric than ours. However its flops and parameters are also greater.

**Self-attention scheme comparison.** We compare Window-based Spectral Self-Attention (WSSA) [38] and spectral TKSA and its variants within the encoder and decoder. We use $'+'$ to signify a

parallel implementation of both attentions (each processing half of the feature channels), and $'-'$ to represent an encoder-decoder implementation. Base-2 is CIDNet-3stg that removes the attention module. Tab. 5 shows the ablation results and LWSA-TKSA yields the most prominent improvement of 1.56dB PSNR compared with Base-2, which shows the effectiveness of our method.

## 4    Conclusion

We present CIDNet, a novel reconstruction framework for CASSI, which leverages a physically motivated chromaticity-intensity decomposition. By disentangling the hyperspectral image into a spatially smooth intensity map and a spectrally informative chromaticity cube, our method enables lighting-invariant reflectance modeling and better preserves spatial-spectral details. We have designed a hybrid spatial-spectral Transformer to recover the sparse and high-frequency chromaticity components and introduced a degradation-aware unfolding strategy with spatially adaptive noise modeling to handle anisotropic noise inherent in the dual-camera CASSI system. Extensive experiments on both simulated and real-world CASSI datasets validate the effectiveness of our approach, achieving state-of-the-art performance in spectral reconstruction and chromaticity fidelity. This work highlights the benefit of physics-aware decomposition and hybrid attention mechanisms in addressing the ill-posed inverse problem of CASSI reconstruction.

## Acknowledgments

This work was supported by National Key R&D Program of China (2024YFF0505603), the National Natural Science Foundation of China (grant number 62271414), Zhejiang Provincial Distinguished Young Scientist Foundation (grant number LR23F010001), Zhejiang 'Pioneer' and 'Leading Goose' R&D Program (grant number 2024SDXHDX0006, 2024C03182), the Key Project of Westlake Institute for Optoelectronics (grant number 2023GD007), the 2023 International Sci-tech Cooperation Projects under the purview of the 'Innovation Yongjiang 2035' Key R&D Program (grant number 2024Z126), National Natural Science Foundation of China (grant number 62501487). (Corresponding author: Xin Yuan.)

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

# A Appendix

In the supplementary material, we provide more details that are not in out main paper:

**(a)** Multispectral Image Formation in Sec. A.1.

**(b)** Proof of Illumination-Invariance of Chromaticity in Sec. A.2.

**(c)** Proof of PAN-Intensity Equivalence in Sec. A.3. We demonstrate the PAN image in dual-camera CASSI is equivalent to the Intensity image.

**(d)** Closed-form Solution of Data-fidelity Term in Sec. A.4.

**(e)** Traditional Optimization-based Methods in Sec. A.5. Some results are visualized.

**(f)** Visual Comparison of HSIs and Chromaticity in Sec. A.6.

**(g)** Noise Map Visualization of DNEM in Sec. A.7.

**(h)** Limitation and Broader Impact of Our Work in Sec. A.8

## A.1 Multispectral Image Formation

Let $(u, v)$ be spatial coordinates and $\lambda \in [\lambda_{\min}, \lambda_{\max}]$ the wavelength. Assume the scene reflectance is $\mathbf{R}(u, v, \lambda)$(this is not the same as data-prior term), and the illumination spectral power is $L(\lambda)$, spatially uniform. Let $s(\lambda)$ be the spectral response function of the grayscale camera, which converts the spectral radiance to a scalar intensity. Then the intensity recorded by the grayscale camera at $(u, v)$ is:

$$\mathbf{I}(u, v) = \int_{\lambda_{\min}}^{\lambda_{\max}} s(\lambda) \cdot L(\lambda) \cdot \mathbf{R}(u, v, \lambda) \, d\lambda. \tag{25}$$

This is the measurement we observe from the grayscale camera. We define a multispectral distribution:

$$\mathbf{X}(u, v, \lambda) = s(\lambda) \cdot L(\lambda) \cdot \mathbf{R}(u, v, \lambda) \tag{26}$$

Then the multispectral chromaticity function is defined as the normalized form of $X(u, v, \lambda)$ over $\lambda$ :

$$\mathbf{C}(u, v, \lambda) = \frac{\mathbf{X}(u, v, \lambda)}{\mathbf{I}(u, v)} = \frac{s(\lambda) \cdot L(\lambda) \cdot \mathbf{R}(u, v, \lambda)}{\int s(\lambda) \cdot L(\lambda) \cdot \mathbf{R}(u, v, \lambda) \, d\lambda}, \tag{27}$$

which leads to the final formulation of spectral product of intensity and chromaticity:

$$\mathbf{X}(u, v, \lambda) = \mathbf{C}(u, v, \lambda) \odot \mathbf{I}(u, v). \tag{28}$$

Next we demonstrate that the chromaticity is illumination invariant, and the intensity can be obtained via a dual-camera setting.

## A.2 Proof of Illumination-Invariance of Chromaticity

Now suppose the illumination changes globally:

$$L(\lambda) \rightarrow \alpha \cdot L(\lambda), \quad \alpha > 0$$

Then:

$$\mathbf{C}'(u, v, \lambda) = \frac{s(\lambda) \cdot \alpha L(\lambda) \cdot \mathbf{R}(u, v, \lambda)}{\int s(\lambda') \cdot \alpha L(\lambda') \cdot \mathbf{R}(u, v, \lambda') \, d\lambda'} \tag{29}$$

$$= \frac{s(\lambda) \cdot L(\lambda) \cdot \mathbf{R}(u, v, \lambda)}{\int s(\lambda') \cdot L(\lambda') \cdot \mathbf{R}(u, v, \lambda') \, d\lambda'} = \mathbf{C}(u, v, \lambda) \tag{30}$$

**Thus,** the chromaticity $\mathbf{C}(u, v, \lambda)$ is invariant to uniform intensity changes in illumination, even when considering the grayscale camera's spectral sensitivity.

## A.3 Proof of PAN-Intensity Equivalence

**Proposition A.1** (PAN-Intensity Equivalence Under Uniform Illumination). *In a dual-camera system comprising a CASSI sensor and a grayscale PAN camera exposed under the same illumination $L(\lambda)$, let $s(\lambda)$ denote the spectral response of the camera. The PAN image $\mathbf{I}_{\mathrm{PAN}}(u, v)$ provides a relative estimate of the scene intensity $\mathbf{I}(u, v)$ defined by the chromaticity-intensity decomposition of the hyperspectral image $\mathbf{X}(u, v, \lambda)$. That is,*

$$\mathbf{I}_{\mathrm{PAN}}(u, v) \approx k \cdot \mathbf{I}(u, v), \tag{31}$$

*where $k$ is a scalar constant that is approximately invariant across spatial coordinates $(u, v)$.*

*Proof.* Our derivation begins with the Retinex theory, which decomposes an image into chromaticity and intensity components. For hyperspectral images, this decomposition is generalized as:

$$\mathbf{X}(u, v, \lambda) = \mathbf{C}(u, v, \lambda) \cdot \mathbf{I}(u, v), \tag{32}$$

where the chromaticity and intensity are defined as:

$$\mathbf{C}(u, v, \lambda) = \frac{\mathbf{X}(u, v, \lambda)}{\int \mathbf{X}(u, v, \lambda')d\lambda'}, \tag{33}$$

$$\mathbf{I}(u, v) = \int \mathbf{X}(u, v, \lambda)d\lambda. \tag{34}$$

In our dual-camera setup with CASSI and PAN sensors under identical illumination $L(\lambda)$, the PAN image formation is modeled as:

$$\mathbf{I}_{\mathrm{PAN}}(u, v) = \int s(\lambda) \cdot L(\lambda) \cdot \mathbf{X}(u, v, \lambda)d\lambda. \tag{35}$$

Substituting the $\mathbf{X}$ into the product of chromaticity and intensity yields:

$$\mathbf{I}_{\mathrm{PAN}}(u, v) = \int s(\lambda)L(\lambda)[\mathbf{C}(u, v, \lambda) \cdot \mathbf{I}(u, v)]d\lambda. \tag{36}$$

Since $\mathbf{I}(u, v)$ is independent of wavelength $\lambda$, it can be factored out of the integral:

$$\mathbf{I}_{\mathrm{PAN}}(u, v) = \mathbf{I}(u, v) \cdot \int s(\lambda)L(\lambda)\mathbf{C}(u, v, \lambda)d\lambda. \tag{37}$$

The key insight is that $\mathbf{C}(u, v, \lambda)$ is normalized ($\int \mathbf{C}(u, v, \lambda')d\lambda' = 1$) and exhibits smooth spectral variation, while $s(\lambda)L(\lambda)$ acts as a broadband low-pass filter. This justifies the approximation:

$$\int s(\lambda)L(\lambda)\mathbf{C}(u, v, \lambda)d\lambda \approx k, \tag{38}$$

where $k$ is a spatial-invariant scalar constant. Therefore, we obtain:

$$\mathbf{I}_{\mathrm{PAN}}(u, v) \approx k \cdot \mathbf{I}(u, v). \tag{39}$$

To address scale ambiguity, we normalize the PAN image to $[0, 1]$ during training and inference, which justifies using PAN as a relative intensity estimate in our chromaticity-intensity framework. $\square$

## A.4 Closed-form Solution of Data-fidelity Term

The **c**-subproblem in Eq. (13) is quadratic and has a closed-form solution:

$$\mathbf{c}^{(k+1)} = \left(\mathbf{H}^\top \mathbf{\Sigma}^{-1}\mathbf{H} + \mu\mathbf{I}\right)^{-1}\left(\mathbf{H}^\top\mathbf{\Sigma}^{-1}\mathbf{y} + \mu\mathbf{z}^{(k)}\right). \tag{40}$$

Note that $\mathbf{H}^\top\mathbf{\Sigma}^{-1}\mathbf{H}$ is a fat matrix and $(\mathbf{H}^\top\mathbf{\Sigma}^{-1}\mathbf{H} + \mu\mathbf{I})^{-1}$ will be difficult to compute and thus we simplify it based on the Sherman-Morrison-Woodbury formula,

$$\left(\mathbf{H}^\top\mathbf{\Sigma}^{-1}\mathbf{H} + \mu\mathbf{I}\right)^{-1} = \mu^{-1}\mathbf{I} - \mu^{-2}\mathbf{H}^\top\left(\mathbf{\Sigma} + \mu^{-1}\mathbf{H}\mathbf{H}^\top\right)^{-1}\mathbf{H}. \tag{41}$$

By plugging Eq. (41) into Eq. (40), we formulate it as

$$\mathbf{c}^{(k+1)} = \frac{\mathbf{H}^\top\mathbf{y} + \mu\mathbf{z}^{(k)}}{\mu} - \frac{\mathbf{H}^\top\left(\mathbf{\Sigma} + \mu^{-1}\mathbf{H}\mathbf{H}^\top\right)^{-1}\mathbf{H}\mathbf{H}^\top\mathbf{y}}{\mu^2} - \frac{\mathbf{H}^\top\left(\mathbf{\Sigma} + \mu^{-1}\mathbf{H}\mathbf{H}^\top\right)^{-1}\mathbf{H}\mathbf{z}^{(k)}}{\mu}. \tag{42}$$

In CASSI systems, $\mathbf{H}\mathbf{H}^\top$ is a diagonal matrix defined as $\mathbf{H}\mathbf{H}^\top \triangleq \mathrm{diag}\{h_1, \ldots, h_n\}$. With $\mathbf{\Sigma} \triangleq \mathrm{diag}(\sigma_1^2, \ldots, \sigma_M^2)$, we obtain:

$$\left(\mathbf{\Sigma} + \mu^{-1}\mathbf{H}\mathbf{H}^\top\right)^{-1} = \mathrm{diag}\left\{\frac{\mu}{\mu\sigma_1^2 + h_1}, \ldots, \frac{\mu}{\mu\sigma_n^2 + h_n}\right\}, \tag{43}$$

$$\left(\mathbf{\Sigma} + \mu^{-1}\mathbf{H}\mathbf{H}^\top\right)^{-1}\mathbf{H}\mathbf{H}^\top = \mathrm{diag}\left\{\frac{\mu h_1}{\mu\sigma_1^2 + h_1}, \ldots, \frac{\mu h_n}{\mu\sigma_n^2 + h_n}\right\}. \tag{44}$$

Let $\mathbf{y} \triangleq [y_1, \ldots, y_n]^\top$ and $[\mathbf{H}\mathbf{c}^{(k)}]_i$ denote the $i$-th element of $\mathbf{H}\mathbf{c}^{(k)}$. We plug Eq. (43) and Eq. (44) into Eq. (42) as

$$\mathbf{c}^{(k+1)} = \mu^{-1}\mathbf{H}^\top\mathbf{y} + \mathbf{c}^{(k)} - \mu^{-1}\mathbf{H}^\top\left[\frac{y_1 h_1 + \mu[\mathbf{H}\mathbf{c}^{(k)}]_1}{\mu\sigma_1^2 + h_1}, \ldots, \frac{y_n h_n + \mu[\mathbf{H}\mathbf{c}^{(k)}]_n}{\mu\sigma_n^2 + h_n}\right]^\top \tag{45}$$

$$= \mathbf{c}^{(k)} + \mathbf{H}^\top\left[\frac{y_1 - [\mathbf{H}\mathbf{c}^{(k)}]_1}{\mu\sigma_1^2 + h_1}, \ldots, \frac{y_n - [\mathbf{H}\mathbf{c}^{(k)}]_n}{\mu\sigma_n^2 + h_n}\right]^\top. \tag{46}$$

Generally this is a generalized form of gradient descent, which is expressed as,

$$\mathbf{c}^{(k+1)} = \mathbf{z}^{(k)} + \mathbf{H}^\top(\mathbf{H}\mathbf{H}^\top + \mu\mathbf{\Sigma})^{-1}(y - \mathbf{H}\mathbf{z}^{(k)}) \tag{47}$$

## A.5 Traditional Optimization-based Methods

We explore two traditional optimization-based paradigms considering a PAN-guided and RGB-guided intensity respectively, where we formulate the optimization problem using TV prior as,

$$\hat{\mathbf{c}} = \mathrm{argmin}_{\mathbf{c}}\, \tfrac{1}{2}||\mathbf{y} - \mathbf{\Phi}(\mathbf{c} \odot \mathbf{i})||_2^2 + \tau\mathbf{TV}(\mathbf{c}), \tag{48}$$

where $\mathbf{\Phi}$ is the sensing matrix determined by the modulation and dispersion process, $\mathbf{c}$ and $\mathbf{i}$ are chromaticity and intensity respectively, the noise estimation term is omitted since it is hard to be estimated in iterative methods. We consider a dual-camera setting where the second camera could be a grayscale or RGB camera, which satisfy $\mathbf{i} = \mathbf{i}^{\mathbf{PAN}}$ $or$ $\mathbf{i}^{\mathbf{RGB}}$. In RGB scenarios, the RGB-guided intensity is a three-channel image, which cannot be multiplied directly with $\mathbf{\Phi}$ due to the channel mismatch. Hence, we interpolate the RGB image to the same spectral channels with corresponding HSIs. Using the HQS framework, Eq. (48) is minimized by solving the following subproblems iteratively by introducing $\mathbf{c} = \mathbf{z}$:

$$\mathbf{c}^{(k+1)} = \mathrm{argmin}_{\mathbf{c}}\, \tfrac{1}{2}||\mathbf{y} - \mathbf{H}\mathbf{c}||_2^2 + \tfrac{\mu}{2}||\mathbf{c} - \mathbf{z}^{(k)}||_2^2, \tag{49}$$

$$\mathbf{z}^{(k+1)} = \mathrm{argmin}_{\mathbf{z}}\, \tfrac{\mu}{2}||\mathbf{z} - \mathbf{c}^{(k+1)}||_2^2 + \tau\mathbf{TV}(\mathbf{z}), \tag{50}$$

Following previous derivation on Eq.49 and traditional TV denoising term Eq.50, we iterate these two steps to approach its finest solution. The reconstruction is compared with iterative methods such as GAP-TV, DeSCI and PIDS and presented in Fig.7. Scene5 is selected for better visulization purpose.

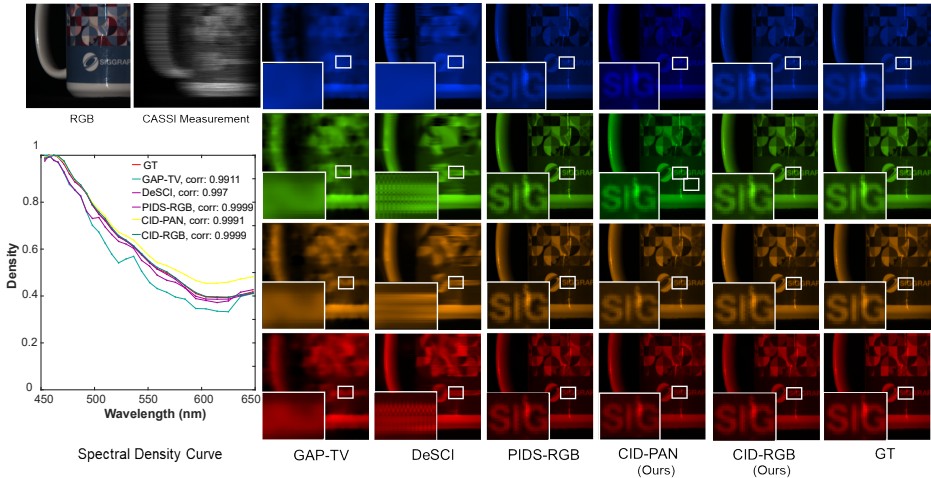

Figure 7: Reconstructed HSIs using traditional optimization-based methods.

## A.6    Visual Comparison of HSIs and Chromaticity

We compare the HSIs and chromaticity using KAIST datasets. As shown in Fig. 9, we select 4 out of 10 test datasets, with the top row being the RGB reference and intensity image. Four spectral images are selected for visulization. It can be seen from the figure that chromaticity enhance the low-light regions while preserving more spatial textures as compared with regular spectral images, demonstrating that chromaticity contains more features.

## A.7    Noise Map

In this paper we propose a dual noise estimation module for data-fidelity term and denoising network. This network module is designed to estimate a spatially adaptive noise map $\mathcal{E}$, two noise map are output corresponding to the gradient projection noise map and the proximal mapping noise map respectively, where we use convolution and channel attention (CA) to enhances informative channels by modeling inter-channel relationships, as shown in Fig. 8. This structure guarantees positivity and adaptiveness of the output noise

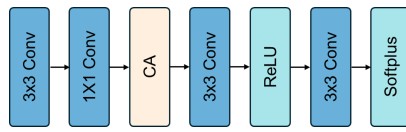

Figure 8: The network structure of step map estimation.

map, suitable for uncertainty modeling or variance-aware image restoration tasks. We conduct the experiment and find in CIDNet-3stg, the gradient projection noise map is prominent in the 1st stage and fades away in the latter 2 stages. While the proximal mapping noise map exhibits finer structure in the 2nd and 3rd stages. This is reasonable since more uncertainty is present in the 1st stage introduced by the noisy measurement while the denosing network focus more on the residual learning.

## A.8    Limitation and Broader Impact of Our Work

Our work assumes a pre-measured intensity and utilizes a grayscale or RGB image obtained by a dual-camera CASSI system to serve as an intensity image. This is how we obtain intensity image and also our limitation. However, we expect that this decomposition framework is applicable to regular CASSI system. By obtaining the intensity image though a regular CASSI training and then freeze this intensity network, continue training the CIDNet for chromaticity reconstruction. This could be further explored in our future work. Moreover, our chromaticity-intensity decomposition framework opens a new paradigm for low-light or shadow-removal hyper-spectral reconstruction since the chromaticity represents more abundant scene/sample information.

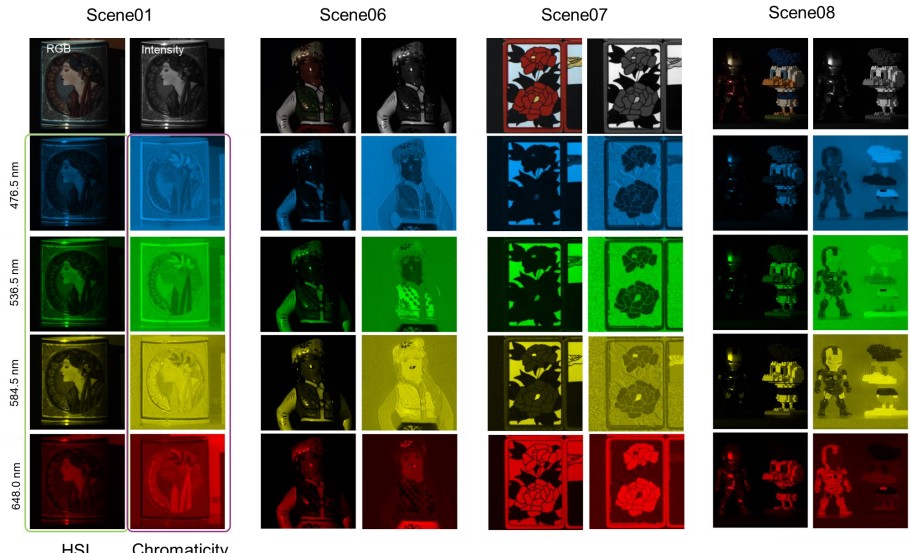

Figure 9: Comparison between HSIs and chromaticity on KAIST test dataset.

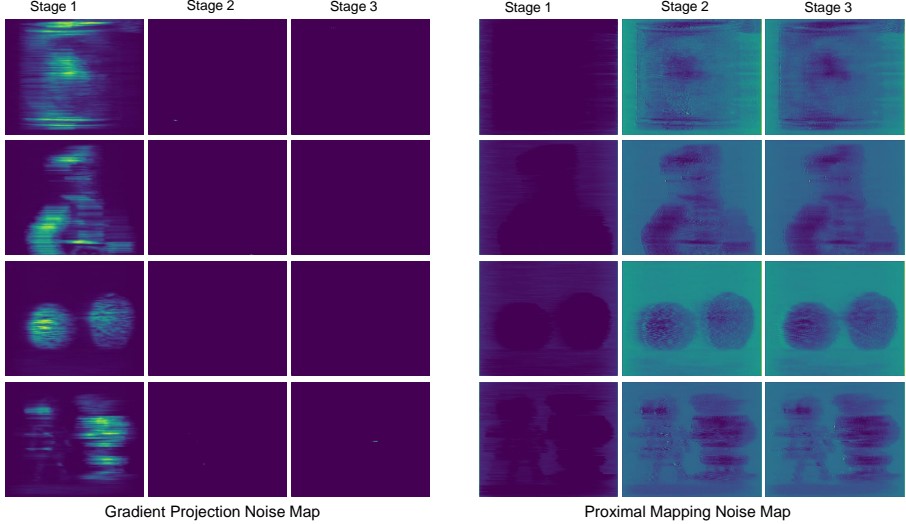

Figure 10: Visulization of dual noise estimation module in CIDNet-3stg.

