# OpenReview forum: "Spectral Compressive Imaging via Chromaticity-Intensity Decomposition"
_NeurIPS.cc/2025/Conference — NeurIPS 2025 poster_

### Official Review · Reviewer_3wmZ · 2025-06-22

**Clarity:** 2
**Significance:** 2
**Originality:** 2
**Rating:** 5
**Confidence:** 4

**Summary:**

This paper describes a novel, interpretable, physically-based method for recovering chromaticity information from compressive spectral measurements taken with the CASSI architecture. Defining hyperspectral chromaticity as the normalized per-pixel spectrum in a hyperspectral image and assuming a known normalizing intensity (e.g., from a second camera), the method modifies the CASSI forward model and poses a regularized optimization problem in chromaticity. This problem is solved with a deep unfolded network to maintain interpretability, and the regularizer is incorporated with alternating optimization. A compact noise estimation model is used to set the noise level for each step to guide the optimization. The regularizer used is a novel hybrid spatial-spectral transformer that preserves high-frequency spatial detail while encouraging sparse local dependencies in the spectral domain. The results show superior performance metrics in chromaticity reconstruction than existing methods on simulated data, and clearer chromaticity reconstruction in one real-world result.

**Questions:**

Please see the weaknesses section. Specifically, please address the following:
- utility of chromaticity in downstream tasks where intensity is also important
- how would real-world intensity image estimation artifacts affect the reconstruction
- metrics in noisy versions of the simulated CASSI data
- the best and worst-performing metrics for the 30 images in the test dataset
- the choice of the ten test images
- best results not highlighted in bold in Table 1

**Ethical Concerns:**

["NO or VERY MINOR ethics concerns only"]

**Final Justification:**

The authors' responses answered my questions satisfactorily, so I will move my rating up.

**Limitations:**

Only one limitation—about the requirement for an intensity image—is described. Please see the weaknesses section for more. Also please move it into the text of the main paper because it is a very significant assumption.

**Quality:**

2

**Strengths And Weaknesses:**

Strengths:
- The method is interpretable via its unfolding structure.
- The HQS approach allows for independent control over the data fidelity and regularization terms.
- The dual-noise estimation framework is interesting in giving the network a map of the uncertainty to tune each unfolded step update.
- In estimating chromaticity, CIDNet seems to perform better than the other existing methods on the 10 images taken for evaluation in simulation and the single real-world image.

Weaknesses:
- The utility of the chromaticity-intensity decomposition is unclear to me. Color information encoded in chromaticity is an important lighting-invariant part of the spectral characteristics of an object. However, the intensity component is important as well for further tasks such as mixed-material analysis, material identification and surgical guidance. So why not reconstruct the hyperspectral cube and then, if needed, decompose it?
- As for the dependence on lighting spectrum and intensity: that can be calibrated out with a spectrally-flat target. For downstream tasks that require intensity, even though reconstructing chromaticity with CIDNet is lighting-independent, getting intensity independent of lighting still requires this calibration.
- Supplement Figure 3 shows the difference between chromaticity and HSI to make the point that chromaticity enhances the signal in the image. However, it also decreases contrast between different materials along the intensity axis, and in real-world settings, would potentially amplify noise in low-signal regions.
- Estimating the intensity image separately has the potential to introduce artifacts—for instance, in occlusion and misalignment in dual-camera setups, and smearing or blurring in the PAN image. How do these affect the reconstruction? Since in the simulated CASSI measurements in the results, the intensity image is taken as the ground truth is not really “dual-camera”.
- Has there been any noise introduced in the simulation to emulate the effects of real-world conditions and get performance metrics?
- There is only one real-world result shown with a dual-camera setup which seems to be a planar scene, therefore not having any of the problems in the dual-camera setup above. Additionally, I do not believe that performing better on one scene adequately validates the real-world superiority of CIDNet.
- Only 10 out of the 30 images in the KAIST dataset were picked for evaluation. How were they picked? Are the average metrics in the last column shown only on these 10? What are the best- and worst-performing metrics for all 30 images? This will help mitigate concerns of selection bias.
- Table 1: In scenes 1, 3, 5, 8, 10, no CIDNet is the best result in either PSNR or SSIM. Please explain why CIDNet is still highlighted as the best result.

Writing fixes:
- Please cite sources for these: (1) line 96, PAN images can be used as intensity, (2) line 130, HQS.
- Here are some suggestions: (1) Figure 1 caption “Chrmaticity”->”Chromaticity”, (2) Figure 2: seems like the labels (b) and (c) are flipped, (3) line 63: “global”->”spatial”, (4) line 127:  unused Mahalanobis norm definition, (5) line 134: “fat matrix”->”wide matrix”, (6) line 247: repeated “the chromaticity”, (7) supplement line 2: “Multispectral”->”Hyperspectral”.
- Please fix the cross-references wherever they say “??”

---

> ### Author Rebuttal · Authors · 2025-07-31
>
> Thanks for your valuable comments, we address your concerns one by one.
>
> > **Q4-1: Utility of chromaticity in downstream tasks where intensity is also important. So why not reconstruct the hyperspectral cube and then, if needed, decompose it?**
>
> A4-1: We agree that both chromaticity and intensity are important for downstream applications such as material identification and surgical guidance. However, in this work, we specifically focus on chromaticity reconstruction, due to the following considerations:
>
> (1) Task difficulty and reconstruction fidelity.
>     Chromaticity encodes fine-grained texture and material information and is generally harder to reconstruct, whereas intensity primarily reflects low-frequency lighting patterns, which are relatively easier to recover. Therefore, we directly supervise and reconstruct the chromaticity component, which proves to be more effective than reconstructing the full hyperspectral cube followed by decomposition (see Table 2). The direct supervision helps preserve high-frequency chromatic information crucial for tasks like object fingerprinting.
>
> (2) Joint ambiguity in decomposition
>     Jointly reconstructing both chromaticity and intensity introduces ambiguity, especially in regions where the product of the two can yield similar measurements but from different chromaticity-intensity combinations. Our current pipeline does not explicitly resolve this ambiguity. However, we observe that a stage-wise strategy—where the first stage reconstructs intensity (with frozen weights) and the subsequent stages focus on chromaticity—still achieves competitive performance. This code will be released.
>
> > **Q4-2: Utility of chromaticity in downstream tasks where intensity is also important, still needs calibration.**
>
> A4-2: We agree that absolute chromaticity and illumination intensity can be obtained through pre-calibration using a spectrally-flat target. In fact, we also plan to incorporate this calibration-based strategy in future work to estimate physically accurate lighting conditions. However, in our current experiments, we adopt a normalized dual-path PAN image as a proxy for intensity. This is not a physically absolute lighting measurement, but a relative intensity estimate. This choice is motivated by two practical considerations:
>
> (1) Stability during training.
>     Using normalized intensity improves data consistency and helps stabilize model training, especially under variations in exposure, gain, and scene content.
>
> (2) Focus on spatial variation rather than absolute magnitude.
>     Our aim is to leverage the spatial structure of intensity changes (e.g., shading, highlights) rather than the absolute illumination levels. The relative intensity still captures rich cues about lighting distribution, which is sufficient for our chromaticity reconstruction pipeline.
>
> > **Q4-3: About noise amplification in dark region.**
>
> A4-3: This is a valid and commonly recognized issue in chromaticity-based representations, especially in low-light or shadowed regions where signal-to-noise ratio is inherently low. Such issue are also extensively discussed in the Retinex and intrinsic image literature (see [1] and [2]). In our case, we agree that this effect exists and is hard to eliminate completely. There are two practical strategies to mitigate it:
>
> (1) Retraining on real-world datasets containing paired chromaticity-intensity hyperspectral data under diverse lighting conditions. This would allow the model to learn better noise-aware priors in low-light regions. Unfortunately, such datasets are currently unavailable to the best of our knowledge.
>
> (2) Post-processing on the reconstructed chromaticity, such as applying TV-denoising or bilateral filtering to suppress artifacts in dark areas. While feasible, this approach introduces additional hyperparameters and lacks ground-truth supervision, making quantitative evaluation difficult.
>
> > **Q4-4: Real-world experiment difficulties due to misalignment.**
>
> A4-4: In our current real-world experiments, however, we use publicly available real CASSI system datasets. That said, we fully acknowledge that in a true dual-camera configuration, such issues can impact the quality of the estimated intensity image, and in turn affect chromaticity reconstruction. We will consider the suggestions.
>
> > **Q4-5: Noise simulation in simulation and real-data condition.**
>
> A4-5: In simulation experiment, both CASSI measurements and the PAN images are noise-free (for fair comparison with the baseline algorithm). In real experiment, follow the setting of TSA-Net [3], we injected 11-bit, quantum efficiency 0.4 shot noise into both CASSI measurements and PAN images.
>
> > **Q4-6: About adding more real-world dataset testing.**
>
> A4-6: We agree that validation on more complex real scenes is important. Due to platform and cost constraints, we have not yet been able to collect a large-scale real dataset. To date, the only publicly available dual‑camera compressive hyperspectral imaging (DCCHI) dataset appears in the paper [4], which provides exactly two real‑scene captures: one ninja scene and one doll scene.
>
> > **Q4-7: About KAIST dataset selection.**
>
> A4-7: The KAIST benchmark for spectral imaging was first proposed in TSA-Net [3], these 10 scenes are randomly chosen, and this benchmark was used in a series of subsequent papers focus on spectral reconstruction like MST[CVPR 2022], DAUHST[NeurIPS 2022], SSR[CVPR 2024], In2SET[CVPR 2024].
>
> > **Q4-8: Table 1 not highlighted best result.**
>
> A4-8: Thank you for pointing this out. We sincerely apologize for the confusion. Few metrics in one baseline work PIDS [5] are incorrect, causing the misunderstanding.  The reconstructed results are available in the original benchmark from [5] (which is a dual-camera CASSI baseline), and we have now revised the table accordingly. By the way, to illustrate the effectiveness of our Chroma-Intensity Decomposition method. We adapted PIDS into PIDS-CIDS, where we formulate the optimizationproblem as：
> $$
>  𝑐̂ = argmin_c (1/2)‖y − Φ(c ⊙ i′)‖² + τ·TV(c − i'),  s.t.  i' = interpolate(i_{rgb})，
> $$
>
> where TV denotes Total Variation and $\text{interpolate}(\mathbf{i}_\text{rgb})$ means to interpolate the RGB image captured by the second camera to spectral channels (28 channels in our case). This is to align with the PIDS setting (the second camera is RGB camera).
>
> | Method      | S1                   | S2                   | S3                   | S4                   | S5                   | S6                   | S7                   | S8                   | S9                   | S10                  | Avg                  |
> | ----------- | -------------------- | -------------------- | -------------------- | -------------------- | -------------------- | -------------------- | -------------------- | -------------------- | -------------------- | -------------------- | -------------------- |
> | PIDS        | 42.09  0.983         | 40.08  0.949         | 41.50  0.968         | 48.55  0.989         | 40.05  0.982         | 39.00  0.974         | 36.63  0.940         | 37.02  0.948         | 38.82  0.953         | 38.64  0.980         | 40.24  0.967         |
> | PIDS-CIDS | 41.02  0.983         | 43.19  0.983         | 40.93  0.930   | 49.23  0.988 | 38.53  0.983         | 39.41  0.978         | 37.49  0.960         | 37.86  0.972         | 39.35  0.965    | 39.06  0.984 | 40.61  0.973         |
>
> It can be found that PIDS-CID better results compared with pure original one. Some visilization results are presented in supplement Figure 7 (denoted as PIDS-RGB).
>
>
> > **Q4-9: Suggestions on Writing fixes.**
>
> A4-9: Thank you for pointing this out. We will correct the writing accordingly.
>
>
>
> **References:**
>
> [1] Wei et al., *Deep Retinex Decomposition for Low-Light Enhancement*, BMVC 2018.
>
> [2] Bell, Sean, et al., *Intrinsic images in the wild*,  ACM TOG, 2014.
>
> [3] Meng, et al., *End-to-end low cost compressive spectral imaging with spatial-spectral self-attention*, ECCV, 2020.
>
> [4] Zhang, et al., *Fast hyperspectral image recovery of dual-camera compressive hyperspectral imaging via non-iterative subspace-based fusion*, IEEE TIP, 2021.
>
> [5] Chen et al., *Prior image guided snapshot compressive spectral imaging*,  IEEE TPAMI 2023.

---

> > ### Comment · Reviewer_3wmZ · 2025-08-05
> >
> > Thank you for your response and for correcting the metrics. Tentatively, looking at the rebuttal, I am inclined to move my rating up. I have one follow-up question:
> >
> > A4-1: The two points discussed here together seem to lead to a problem. (1) says that if there are high-frequency patterns, CIDNet will recover them in chromaticity. (2) says that CIDNet does not explicitly resolve the ambiguity–what stops CIDNet from creating artificial patterns that are still consistent with the measurement?
> >
> > In general, I believe that more results with real data need to be shown to make a compelling case for CIDNet, but I acknowledge the difficulty in implementation and acquisition of such data.

---

> > > ### Author Response · Authors · 2025-08-05
> > >
> > > We sincerely thank the reviewer for the positive feedback and for considering an improved rating. Regarding the concerns on ambiguity, we response step-by-step:
> > >
> > > > **(1) “CIDNet will recover high-frequency patterns in chromaticity.”**
> > >
> > > Yes, that is correct. Our CIDNet is designed to reconstruct high-frequency chromaticity details by leveraging the intensity information provided by the secondary camera. In our dual-path setup, the intensity component is treated as a known input and injected into the network. The network is supervised to predict the chromaticity map, which encodes fine-grained material and texture information. By decoupling the intensity as external guidance, CIDNet focuses solely on learning chromaticity that are not captured by intensity, ensuring high-frequency patterns are accurately recovered in the chromaticity domain.
> > >
> > > > **(2) “CIDNet does not explicitly resolve the ambiguity.”**
> > >
> > > CIDNet is specifically designed for chromaticity reconstruction conditioned on a known intensity input. If both chromaticity and intensity are unknown and need to be jointly estimated, the decomposition inevitably becomes ambiguous. This ambiguity arises because, without a known intensity, there are infinitely many chromaticity-intensity pairs that can produce the same measurement. This is a common issue in inverse problems; for instance, a measurement value of 4 could result from a product of 1×4, 2×2, or infinitely many chromaticity-intensity pairs.
> > >
> > > However, in our method, this ambiguity is significantly reduced because the intensity is directly obtained from the second camera. By conditioning the chromaticity reconstruction on this known intensity, CIDNet effectively eliminates most of the ambiguity space.
> > >
> > > > **(3) what stops CIDNet from creating artificial patterns that are still consistent with the measurement?**
> > >
> > > We appreciate the reviewer’s critical insight. We address this concern from two perspectives:
> > >
> > > **First**, the neural networks tend to prioritize learning low-frequency signals (see [1]), while high-frequency components such as fine-grained chromaticity details are inherently more difficult to model. Although supervision with high-frequency chromaticity labels mitigates this issue, it does not fully eliminate the risk of the network hallucinating artificial patterns that remain consistent with the measurements. This challenge is typical in inverse problems where fine details are weakly constrained by the measurement model alone.
> > >
> > > **Second**, the ambiguity becomes more pronounced if the network attempts to jointly learn both high-frequency chromaticity and low-frequency intensity simultaneously. This introduces several practical challenges: How should the loss functions be designed to balance the reconstruction of chromaticity and intensity? How to design network modules that distinctly handle chromaticity and intensity learning without interference? Supervising chromaticity, intensity, and full spectral reconstruction simultaneously often leads to unstable training, a phenomenon frequently observed in low-light enhancement tasks (see [2]).
> > >
> > > While we don't have a formal solution for joint estimation ambiguity, we observe that a sequential strategy (first stage for intensity, remaining stages for chromaticity) shows promise, though not matching our dual-path performance. We plan to investigate this further in future work.
> > >
> > > **References:**
> > >
> > >
> > > [1] Rahaman et al., *On the Spectral Bias of Neural Networks*, ICML, 2019.
> > >
> > > [2] Wu et al., *Interpretable optimization-inspired unfolding network for low-light image enhancement*, TPAMI, 2025.

---

> > > > ### Comment · Reviewer_3wmZ · 2025-08-07
> > > >
> > > > Thank you for your responses. They resolve my questions, so I will move my rating up.

---

### Official Review · Reviewer_DTCy · 2025-06-28

**Clarity:** 2
**Significance:** 2
**Originality:** 2
**Rating:** 4
**Confidence:** 4

**Summary:**

The paper proposes a new method called CIDNet, which revolutionizes dual-camera CASSI reconstruction by splitting images into intensity and chromaticity components, enabling robust spectral recovery under varying light. Its hybrid Transformer architecture captures rich textures and sparse spectral features, while adaptive noise modeling handles complex degradation.

**Questions:**

see weakness.

**Ethical Concerns:**

["NO or VERY MINOR ethics concerns only"]

**Final Justification:**

The author's reply solved some problems, so I will adjust the score. However, it is not appropriate to compare the methods of different camera systems, so I can still only give the borderline score.

**Limitations:**

yes

**Quality:**

2

**Strengths And Weaknesses:**

Strengths:

1. The chromaticity-intensity decomposition leverages the physical properties of light and reflectance, leading to a more robust and interpretable reconstruction.

2. The experiments demonstrate that CIDNet achieves good performance.

Weaknesses:

1. The authors claimed that the intensity image can be approximated as a PAN image (line 96) without specific explanations. Is there any theory that can support this point?

2. This work integrates some network modules from other works, such as top K attention, noise estimation, etc. The contributions of these are very limited but have brought almost all the improvements to the network. The core of this work is the estimation of Chromaticity-Intensity. By contrast, the gain it brings is very limited.

3. The mark that appears after the word 'appendix', for example, lines 62 and 101. Please carefully check the manuscript and correct similar mistakes.

---

> ### Author Rebuttal · Authors · 2025-07-31
>
> Thanks for your valuable comments, we address your concerns one by one.
>
> > **Q3-1: On the justification of using PAN image as intensity.**
>
> A3-1: The idea of using PAN image as intensity is inspired by *Retinex theory* [1] widely used for low-light image enhancement, where a RGB image is decomposesed into reflectance and illumination. This is typically formulated as:
> $$
> \mathbf{X}(u,v) = \mathbf{C}(u,v) \cdot \mathbf{I}(u,v)
> $$
> where $\mathbf{C}$ denotes chromaticity (color) and $\mathbf{I}$ denotes intensity (illumination). We generalize this concept to hyperspectral images as:
> $$
> \mathbf{X}(u,v,\lambda) = \mathbf{C}(u,v,\lambda) \cdot \mathbf{I}(u,v),
> $$
> where chromaticity and intensity are defined as:
> $$
> \mathbf{C}(u,v,\lambda) = \frac{\mathbf{X}(u,v,\lambda)}{\int \mathbf{X}(u,v,\lambda') d\lambda'}, \quad
> \mathbf{I}(u,v) = \int \mathbf{X}(u,v,\lambda) d\lambda.
> $$
> In a dual-camera setup with a CASSI sensor and a grayscale PAN camera, both exposed under the same illumination $ L(\lambda) $, $s(\lambda)$ is the spectral camera response. we model the PAN image as the integral of three terms:
>
> $$
> \mathbf{I}_{\text{PAN}}(u,v) = \int s(\lambda) \cdot L(\lambda) \cdot \mathbf{X}(u,v,\lambda) \cdot d\lambda.
> $$
>
> Substituting the decomposition of $ \mathbf{X} $ , we obtain:
>
> $$
> \mathbf{I}_{\text{PAN}}(u,v) = \mathbf{I}(u,v) \cdot \int s(\lambda) L(\lambda) \mathbf{C}(u,v,\lambda) \, d\lambda.
> $$
>
> The key observation is that $\mathbf{C}(u,v,\lambda)$ is normalized (the integral of $\mathbf{C}(u,v,\lambda)$ over wavelength equals 1) and typically smooth or slowly varying in $\lambda$, while $s(\lambda)L(\lambda)$ acts as a broadband low-pass kernel. This enables the approximation:
> $$
> \int s(\lambda) L(\lambda) \mathbf{C}(u,v,\lambda) d\lambda \approx k,
> $$
> where $k$ is a scalar constant across spatial positions. Thus, we have:
> $$
> \mathbf{I}_{\text{PAN}}(u,v) \approx k \cdot \mathbf{I}(u,v).
> $$
>
> To resolve the scale ambiguity, we normalize PAN to [0,1] during both training and inference. This justifies using PAN as a relative intensity estimate in our chromaticity–intensity framework, and we will add this proof into our supplement.
>
> > **Q3-2: Concerns on network module design and the gain of chromatic-intensity decomposition.**
>
> A3-2:  **This decomposition is the core contribution** of our paper and brings several conceptual and practical advantages that go beyond simple numerical gains in PSNR or SSIM. We demonstrate this in two aspects:
>
> (1) This decomposition allows us to separate illumination material-dependent reflectance (chromaticity) and intensity from single measurement, providing a representation that is more physically meaningful and invariant to lighting changes. We believe this will significantly benefit CASSI-based reconstruction and its downstream applications such as remote sensing, water quality detection, or hyperspectral material classification, where chromaticity serves as a more precise fingerprint of material properties.
>
> (2) Secondly, by explicitly modeling intensity, we enable applications that target illumination manipulation. For instance, we can apply gamma correction to the intensity map to achieve low-light enhancement, which would be infeasible if illumination and reflectance were entangled.
>
> About the network design, although our network components (e.g., spatial–spectral transformers) are not entirely novel, their design is carefully adapted to suit the properties of chromaticity (sparse spectra and high-frequency details). We also propose a dual noise estimation module to handle anisotropic noise across different reconstruction stages. This is backed by theoretical modeling and empirical evidence, and improves convergence and visual quality.
>
> The gains of this decomposition is our main contribution, which is manifest in two aspects: spectral reconstruction (Table 1) and chromaticity reconstruction (Table 2). Ours outperform single-camera (DAUHST[NeurIPS 2022], SSR[CVPR 2024]) and dual-camera schemes (PIDS[TPAMI 2023], In2SET[CVPR 2024]).
>
> In summary, our goal is not only to improve metrics but to reformulate the problem in a more interpretable and physically grounded way, which we believe is a valuable step for the CASSI research community.
>
> > **Q3-3: Manuscript check.**
> >
> A3-3: Thanks for the suggestion, we will correct the errors accordingly.
>
>
> **References:**
>
> [1] Wei et al., *Deep Retinex Decomposition for Low-Light Enhancement*, BMVC 2018.

---

> > ### Comment · Reviewer_DTCy · 2025-08-02
> >
> > Thanks for the authors' reply. I still keep the questions as follows:
> >
> > 1. I think that considering spectral snapshot reconstruction from the view of low-light image enhancement is feasible in some dark scenes, but not suitable for all scenes. Although current work can achieve SOTA performance in dual camera snapshot datasets, its gains mostly come from the incremental improvement of existing methods. I speculate that the current datasets may not be good for demonstrating the proposed method. If possible, it might be more appropriate to construct some low-light snapshot imaging datasets to test the method.
> >
> > 2. The 'Retinex theory' is derived from the RGB image. Whether it supports the spectral image or not is unknown.
> >
> > 3. The claimed method could perform better than single-camera methods, but I do not see any experiments about the single-camera system. It is not fair to directly compare the methods in different imaging systems. Also, may I ask, single-camera is the more mainstream snapshot imaging mode. Why not conduct experiments on this?

---

> > > ### Author Response · Authors · 2025-08-05
> > >
> > > We thank the reviewer for your valuable comments. Below are our point-by-point responses.
> > >
> > > > **Q1.1:  The feasibility in some dark scenes, but being not suitable for all scenes.**
> > >
> > > A1.1. Our chromaticity-intensity decomposition introduces Retinex-based theory into hyperspectral CASSI reconstruction for the first time. It should be noted that our method and experimental validation take aim at regular-light scenes instead of low-light scenes (as validated in Manuscript Fig. 3-4).
> > >
> > > While inspired by low-light framework, we made key technical modification to extend retinex theory from low-light scenes to regular-light scenes. Low-light methods [1] use the maximum value across RGB channels  $\text{max}_{c\in{R,G,B}}\{X^{c}\}$ to amplify dark illumination, while we use the mean value across spectral channels  to preserve natural intensity in regular scenes.
> > >
> > > To improve the low-light reconstruction, changing the intensity as the maximum of spectral channels $\text{max}_{c\in{R,G,B}}\{X^{c}\}$ is a potential solution but is out of the scope of this work. Thanks for your question again, we will study low-light CASSI reconstruction in the future.
> > >
> > > > **Q1.2:  Although current work can achieve SOTA performance in dual camera snapshot datasets, its gains mostly come from the incremental improvement of existing methods.**
> > >
> > > A1.2.  We would like to highlight that our contributions are two-fold:
> > >
> > > (1) We are the first to introduce chromaticity decomposition into the CASSI reconstruction framework. By learning the chromaticity, which is a lighting-invariant representation of the intrinsic material properties, we can better capture the essential spectral information of the scene. This is a novel approach in the spectral compressive imaging domain, as no prior work has considered the lighting-and-material decomposition problem in this context.
> > >
> > > (2) Secondly, we reformulate chromaticity learning into a model-based method by redesigning the network for spectral sparsity and high-frequency details, and introducing a learnable spatially-varying degradation model, which generalizes first-order data-consistency updates where we use a learnablel $\mathbf{\Sigma}_{\theta}$ to represent the spatially-varying noise. Forward model is $y = \Phi x$.
> > >
> > > | Method | Data-consistency |
> > > | ------  | ---------------------- |
> > > | ISTA [CVPR 2018]    | $x = z + \Phi^\top(y - \Phi z)$ |
> > > | GAP [IJCV 2023]   | $x = z + \Phi^\top (\Phi \Phi^\top)^{-1} (y - \Phi z)$ |
> > > | HQS [NeurIPS 2022]    | $x = z + \Phi^\top (\Phi\Phi^\top+\mu \mathbf{I})^{-1} (y-\Phi z)$ |
> > > | Ours   | $x= z + \Phi^\top\mathbf{\Sigma}_{\theta}(y - \Phi z)$ |
> > >
> > >
> > > > **Q1.3. I speculate that the current datasets may not be good for demonstrating the proposed method.**
> > >
> > > A1.3. Thanks for your suggestion. To further verify the effectiveness of our method, the proposed CIDNet is tested in another spectral dataset (ICVL NTIRE [3]). In the same dual-camera setting, it is clear that our method generaize well  and still is SOTA. We will add such results in final paper.
> > >
> > > |     Model      |  PSNR & SSIM   |
> > > | ---------------  | ----------- |
> > > | MST-L-DC   |39.39, 0.982 |
> > > | GAP-net-9stg-DC  | 39.09, 0.977 |
> > > | DAUHST-3stg-DC  | 42.33, 0.985 |
> > > | CIDNet-3stg    | 43.34, 0.992 |
> > >
> > > > **Q1.4. If possible, it might be more appropriate to construct some low-light snapshot imaging datasets to test the method.**
> > >
> > > A1.4. We thank the reviewer's suggestion. Unfortunately, there are no public low-light spectral datasets available until now. Collecting such a dataset will need huge labor works and thus it is impractical in such a short rebuttal period.  We will consider doing it in the future works.
> > >
> > >
> > > **References:**
> > >
> > > [1] Wu et al., *Interpretable optimization-inspired unfolding network for low-light image enhancement*, TPAMI, 2025.

---

> > > > ### Author Response · Authors · 2025-08-05
> > > >
> > > > > **Q2. Whether 'Retinex theory'  supports the spectral image or not is unknown.**
> > > >
> > > > A2. According to Grahn and Geladi [2], spectral decomposition provides more accurate optical information than RGB, as RGB uses only a 3-channel R-G-G-B filter, while our method captures the full spectral range (400–600nm, 28 bands), making it a more precise and reasonable approach.
> > > >
> > > > > **Q3. The claimed method could perform better than single-camera methods, but I do not see experiments about the single-camera system. It is not fair to directly compare the methods in different imaging systems. Also, may I ask, single-camera is the more mainstream snapshot imaging mode. Why not conduct experiments on this?**
> > > >
> > > > A3. We agree that dual-camera CASSI system has the advantages over single-camera one as demonstrated in all previous dual-camera CASSI works. We illustrate this from two aspects: simulation and real-data.
> > > >
> > > > (1) For simulation section, we compared our method with single-camera and dual-camera methods (Tab1 and Tab 2).
> > > > For dual-camera baselines, we share the same setting except the network.
> > > >
> > > > (2) For real-data experiments, we only compare with dual-camera methods because our approach inherently relies on the additional intensity image from the second camera to guide chromaticity reconstruction. Comparing with single-camera systems would be unfair, as the extra spatial information from the dual-path setup provides an extra spatial information.
> > > >
> > > > We appreciate this valuable suggestion.
> > > >
> > > > **References:**
> > > >
> > > > [2] Garini, et al., *Spectral imaging: principles and applications*, Applied Optics, 2007.
> > > >
> > > > [3] *NTIRE 2022 Spectral Recovery Challenge and Data Set*. CVPRW 2022.

---

> > > ### Author Response · Authors · 2025-08-08
> > >
> > > Dear Reviewer DTCy,
> > >
> > > Thank you for your valuable comments that has helped improve our manuscript. As the author-reviewer discussion deadline approaches, we would appreciate the opportunity to continue our dialogue. We have carefully considered your comments and provided detailed responses to address your concerns. We would be grateful to know if our clarifications have been helpful and whether there are any remaining points you would like us to address.
> > >
> > > Thank you for your time and consideration.
> > >
> > > Sincerely,
> > > Authors

---

### Official Review · Reviewer_Ka4a · 2025-07-01

**Clarity:** 2
**Significance:** 2
**Originality:** 2
**Rating:** 4
**Confidence:** 5

**Summary:**

The paper proposes CIDNet for dual-camera compressive hyperspectral imaging. It factors the HSI cube into a panchromatic-derived intensity map and a chromaticity tensor, then embeds the intensity into the forward model via an intensity-weighted CASSI mask. Reconstruction is carried out by a nine-stage unfolded solver that alternates physics-based gradient steps with an asymmetric hybrid transformer, while a dual noise-estimation block adaptively tunes both the gradient term and the denoiser. On CAVE, KAIST, and real data, CIDNet surpasses recent transformer baselines in PSNR/SSIM with fewer parameters, and ablations confirm the benefit of each module.

**Questions:**

1.The paper criticises prior work for conflating illumination with reflectance, yet in a dual-camera setting the PAN and CASSI sensors run independent auto-exposure, so differences in shutter time and ISO mean their irradiances are on distinct scales. Because the PAN gray image is used verbatim as the intensity map I(x,y) in the chromaticity–intensity decomposition—without any radiometric calibration or learnable scale factor—might the resulting exposure mismatch simply be absorbed into the chromaticity term C(x,y,λ), thereby recreating the very illumination-reflectance entanglement the manuscript seeks to avoid?

2.In the Dual Noise-Estimation Module (Eqs. 19–20) a single lightweight CNN outputs both the gradient-projection covariance Σ(k) and the denoising weight ω(k), yet the manuscript neither (i) explains how Σ(k) is constrained to remain positive-definite nor (ii) analyses possible gradient interference between the two predictions. Could the authors therefore clarify (1) which activation or re-parameterisation (e.g., soft-plus, exponential) is used to ensure every entry of Σ(k) is strictly positive and the associated matrix inversion remains numerically stable, and (2) whether the norms/variances of Σ(k) and ω(k) remain well-behaved throughout training; additionally, if the two quantities are predicted by separate branches or one is fixed (e.g., Σ(k)=σ²I), how do PSNR/SSIM and convergence speed respond?

3.The paper mainly compares CIDNet with single-camera baselines such as DeSCI and MST-L; to make the evaluation complete and fair, please include (i) those single-camera models adapted to the dual-camera setting and/or (ii) additional methods originally des.

4.The manuscript claims that Chromaticity-Intensity decomposition outperforms plain end-to-end 3-D convolution, yet Table 3 compares only the Base-1 model (no attention) with versions that add the proposed modules. Please include an additional baseline that directly predicts the HSI cube without C-I decomposition while still using HSST and DNEM, so we can verify that the reported gains are attributable to the decomposition itself rather than to the attention architecture.

**Ethical Concerns:**

["NO or VERY MINOR ethics concerns only"]

**Limitations:**

Yes

**Quality:**

3

**Strengths And Weaknesses:**

Strengths: the method integrates a physically motivated chromaticity-intensity decomposition and an intensity-weighted mask, couples them with a well-structured nine-stage unfolded solver, introduces an efficient hybrid attention (window-spatial + sparse spectral) backbone, and adds a dual noise-estimation block, yielding state-of-the-art PSNR/SSIM on CAVE, KAIST and two real scenes with fewer parameters than recent transformer baselines.
Weaknesses: the PAN-derived intensity prior is used without exposure/ISO calibration, the jointly regressed Σ and ω lack positive-definite constraints and stability evidence, real-scene evaluation is minimal, strong dual-camera baselines are absent, ablations are coarse (Top-K size, stage sharing, fixed vs. learned Σ/ω, etc.).

---

> ### Author Rebuttal · Authors · 2025-07-31
>
> Thanks for your recognition and valuable comments, we address your concerns one by one.
>
> > **Q2-1: Exposure mismatch in dual-camera chromaticity-intensity decomposition.**
>
> A2-1: This is a valid concern. However, in practice, we apply normalization to the intensity map to mitigate exposure differences both during training and testing. During training (simulated data), the intensity map is normalized by averaging across all spectral channels: $I(x, y) = \frac{1}{L} \sum_{\lambda=1}^L X(x, y, \lambda)$. During real testing (dual-camera system), we normalize the captured PAN image by dividing by its maximum value: $I_{\text{norm}}(x, y) = \frac{I_{\text{PAN}}(x, y)}{\max(I_{\text{PAN}})}$.
>
> As a result, both the intensity and chromaticity maps are treated as relative quantities. While the absolute radiometric scale may be lost, the relative spatial distribution of intensity is preserved, which is sufficient to support meaningful chromaticity estimation and reconstruction. Moreover this formulation is widely accepted in related literature on (RGB) low-light imaging and intrinsic image decomposition, where illumination is often treated as a relative quantity and gamma correction is used to simulate different lighting conditions (see [1,2]). For example, we can enhance the perceived brightness via: $I_{\text{gamma}}(x, y) = I_{\text{norm}}(x, y)^{\gamma}, \quad \gamma \in (0.5, 2.0)$.
>
>
>
> > **Q2-2: Concerns about Dual Noise-Estimation Module (DNEM).**
>
> A2-2: We thank the reviewer for the interest in Dual Noise-Estimation Module (DNEM). We explain the concerns in two aspects.
>
> (1) Positive-definiteness of $\Sigma^{(k)}$.
>     We argue that $\Sigma^{(k)}$ is not a full covariance matrix but a *diagonal positive definite matrix* representing spatially varying noise variance (see [3]). Specifically, $\Sigma^{(k)} = \operatorname{diag}(\sigma_1^{2}, \dots, \sigma_M^{2})$. To ensure all entries are strictly positive and avoid numerical instability during inversion, we apply the Softplus activation to the output of the CNN branch predicting log-variances and add a small constant $\varepsilon$ for numerical stability: $\sigma_i^{2} = \log(1 + \exp(\hat{\sigma}_i)) + \varepsilon$. where $\hat{\sigma}_i$ is the raw output from the network. This guarantees each entry in $\Sigma^{(k)}$ remains positive-definite by design.
>
> (2) Gradient interference and stability analysis.
>     We employ a shared CNN encoder, followed by two lightweight separate heads to predict $\Sigma^{(k)}$ and ω(k), respectively. This modular design alleviates potential gradient interference. Results (Table 3 and SM Figure 4) show that using learned $\Sigma^{(k)}$ improves reconstruction PSNR and SSIM, highlighting the importance of noise-adaptive spatial modeling in CASSI. We will revise the manuscript to clearly state that $\Sigma^{(k)}$ is a diagonal variance map and provide details about softplus activation and the dual-head structure to improve clarity.
>
> > **Q2-3: Single-camera models adapted to the dual-camera setting.**
>
> A2-3: In the original submission, we compared our proposed CIDNet with In2SET (CVPR-2024) and PIDS (TPAMI-2023), which is the state-of-the-art method for dual-camera compressive hyperspectral reconstruction. Our results showed that CIDNet outperforms In2SET and PIDS significantly in simulated and real scenarios. To futher demonstrate the effectivesss of our decomposition method, we have also adapted popular single-camera reconstruction models (unfolding framework GAP-Net [4] and an E2E model MST [5]) to the dual-camera setting. The detailed results are now summarized as follows. The expriments are compared with 9-stage unfolding framework and it is shown that with Chroma-Intensity Decomposition greatly improve the reconstruction quality.
>
> |              | GAP-net-9stg | GAP-net-9stg w/ CI | MST-L | MST-L w/ CI | CID-Net-9stg w/ CI|
> | ------------- | ------------ | --------------- | ----- | -------- | ------------ |
> | Avg PSNR (dB) | 33.26        | 38.62           | 35.18 | 39.22    | 44.12    |
> | Avg SSIM      | 0.917        | 0.977           | 0.948 | 0.980    | 0.991    |
>
> > **Q2-4: Additional baseline that directly predicts the HSI cube without Chro-Intensity Decomposition.**
>
> A2-4: We have added a new baseline that directly predicts the HSI cube without Chro-Intensity Decomposition. The performance of CIDNet improved significantly after adding C-I decomposition. We compared the MST [5] with our CID-Net. At the same time, without C-I decomposition (i.e., direct reconstructing hyperspectral images without input PAN images), our network still has better performance with lower parameters and FLOPs.
>
> |               | MST-S | CID-Net-3stg w/o CI | CID-Net-3stg w/ CI |
> | ------------- | ----- | -------------------- | ------------------ |
> | Avg PSNR (dB) | 34.26  | 36.85                | 42.51          |
> | Avg SSIM      | 0.935  | 0.961                | 0.980          |
> | Params (M)    | 2.03    | 1.40           | 1.40        |
> | FLOPs (G)     | 28.15  | 24.80            | 24.80         |
>
> **References:**
>
> [1] Chen, Chen, et al. "Learning to See in the Dark." *CVPR*, 2018.
>
> [2] Bell, Sean, et al. "Intrinsic images in the wild." *ACM TOG*, 2014.
>
> [3] Glaubitz, Jan, et al. "Generalized sparse Bayesian learning and application to image reconstruction." SIAM/ASA Journal on Uncertainty Quantification 11.1 (2023): 262-284.
>
> [4] Meng, et al. "Gap-net for snapshot compressive imaging".  *arXiv *, 2020.
>
> [5] Cai, et al. "Mask-guided spectral-wise transformer for efficient hyperspectral image reconstruction[C]//Proceedings of the IEEE/CVF conference on computer vision and pattern recognition".  *CVPR*, 2022.

---

### Official Review · Reviewer_M6fb · 2025-07-01

**Clarity:** 1
**Significance:** 3
**Originality:** 3
**Rating:** 4
**Confidence:** 5

**Summary:**

The manuscript proposed a solution for CASSI image reconstruction.
Main novelties could include:
1. Separates the chromaticity-intensity map of an image
2. Models the noise covariance and denoising strength with a neural network
3. Top-K spectral attention for reducing the computational overhead.

**Questions:**

Is prox and proj implemented as the proposed transformers?

**Ethical Concerns:**

["NO or VERY MINOR ethics concerns only"]

**Final Justification:**

The authors have answered my questions, hence I will keep my previous positive recommendation.

**Limitations:**

Yes

**Paper Formatting Concerns:**

Fig. 2 caption is not in the correct order.

**Quality:**

3

**Strengths And Weaknesses:**

Strength:
1. A dedicated network structure that reaches SATO in the datasets.
2. Cascade network architectures that realize the functionality of an optimization scheme.
3. The plots and equations are generally informative.

Weakness:
1. Some equations are confusing; the positive/negative signs and the brackets in Eqs 10, 11, 12 need to be checked.
2. LWSA TKSA are not explained, and it is not clear what has been compared in the ablation.

It is generally good work, I will accept the publication once the authors clarify the unclear points and correct the figures and formulas in the manuscript.

---

> ### Author Rebuttal · Authors · 2025-07-31
>
> Thanks for your recognition and valuable comments, we address your concerns one by one.
>
> > **Q1-1: Some equations are confusing, signs and brackets need to be checked.**
>
> A1-1: Thanks for your kindly suggestions. We will carefully correct the writting errors in the final manuscript, especially your mentioned
>
> $\hat{\mathbf{c}} = \arg min_{\mathbf{c}} \left(-\frac{1}{2} (\mathbf{y} - \mathbf{H}\mathbf{c})^\top \boldsymbol{\Sigma}^{-1} (\mathbf{y} - \mathbf{H}\mathbf{c})\right) + \tau R(\mathbf{c})$
>
> $\textstyle \hat{\mathbf{c}} = \arg min_{\mathbf{c}, \mathbf{z}} \left(-\frac{1}{2} (\mathbf{y} - \mathbf{H}\mathbf{c})^\top \boldsymbol{\Sigma}^{-1} (\mathbf{y} - \mathbf{H}\mathbf{c})\right) + \tau R(\mathbf{z}), \quad \text{s.t.} \quad \mathbf{c} = \mathbf{z}. $
>
> would be corrected as
>
> $\hat{\mathbf{c}} = \arg min_{\mathbf{c}} \left(\frac{1}{2} (\mathbf{y} - \mathbf{H}\mathbf{c})^\top \boldsymbol{\Sigma}^{-1} (\mathbf{y} - \mathbf{H}\mathbf{c})\right) + \tau R(\mathbf{c})$
>
> $\textstyle \mathbf{\hat{c}},\mathbf{\hat{z}}  = \arg min_{\mathbf{c}, \mathbf{z}} \left(-\frac{1}{2} (\mathbf{y} - \mathbf{H}\mathbf{c})^\top \boldsymbol{\Sigma}^{-1} (\mathbf{y} - \mathbf{H}\mathbf{c})\right) + \tau R(\mathbf{z}), \quad \text{s.t.} \quad \mathbf{c} = \mathbf{z}.$
>
> > **Q1-2: LWSA and TKSA are not explained.**
>
> A1-2: We provide motivation and technical details of our hybrid transformer module: LWSA (Local Window Spatial Attention) and TKSA (Top-K Spectral Attention).
>
> *Motivation*:
> Chromaticity exhibits rich high-frequency details in the spatial domain (see Figure 1), but its spectral signatures tend to be sparse and low-rank (see Figure 5). To effectively model this structure, we use LWSA (a window-based spatial attention) to capture local spatial textures. We design a Top-K Spectral Attention (TKSA) to focus on the most informative spectral dimensions for each spatial location. While LWSA is a typical window transformer, we focus on the explanation of TKSA.
>
> *Techinical details of TKSA*:
> Top-K Spectral Attention (TKSA): Instead of applying global attention across all spectral bands—which is both computationally expensive and often dominated by noisy or redundant information—we propose to:
> - Divide the image into non-overlapping spatial windows.
> - For each window, we first calculate the attention map in the spectral dimension, this attention map size is C×C, for each line in the attention map, we keep the maximum K value, and the other values are set to -∞, so that after the softmax operation, these values will become 0, and we finally use this attention map to weight the key and get the output.
>
> > **Q1-3: Ablation study explanation.**
>
> A1-3:  Below, we clarify the motivation and significance of each experiment by highlighting the core novelties of our work. Our main contribution is to reformulate spectral reconstruction as a chromaticity prediction problem by decomposing the spectral image into intensity × chromaticity. The intensity is directly obtained from the PAN image captured by the dual-camera CASSI system, while chromaticity contains the essential spectral variation. Based on this, we design targeted modules for effective chromaticity reconstruction.
>
> - **Table 3** presents the contribution of three components:
>
> (1) Int.: our decomposition-based framework that incorporates PAN-guided intensity;
>
> (2) HSST: a hybrid spatial-spectral transformer tailored for local spatial and sparse spectral patterns in chromaticity;
>
> (3) DNEM: a dual-noise estimation module that models spatially-varying degradation across unfolding stages.
>
> - **Table 4** compares different attention types within the encoder-decoder. The hybrid spatial-spectral attention (TopK) yields the best performance, validating the importance of our design.
> - **Table 5** further verifies that our intensity-chromaticity decomposition generalizes well across different types of frameworks (iterative, end-to-end, and unfolding), demonstrating its broad applicability and core novelty.
>
> > **Q1-4: Is prox and proj implemented as the proposed transformers?**
>
> Unfolding network typically alternates between a gradient projection and a proximal mapping step to enforce measurement consistency and data prior updating respectively. Hereby we use $\mathbf{proj}$ and $\mathbf{prox}$ to denote the gradient projection and proximal mapping respectively, where $\mathbf{proj}$ is some linear operators and $\mathbf{prox}$ contains a learnable network with our hybrid transformer.

---

### Note · Authors · 2025-08-13

# Response to All Reviewers and Area Chairs

We thank all reviewers and area chairs for their time and valuable comments. All source code and pre-trained models will be made publicly available for further research. After discussing with reviewers and providing more clarifications/results/analyses, we would like to give a brief response.

(a) Reviewer M6fb and Ka4a hold a $\text{positive}$ side for our work.

**Expert reviewers M6fb and Ka4a both provided positive evaluations**. Reviewer M6fb praised our technical innovations including chromaticity-intensity separation, neural network-based noise modeling, and top-K spectral attention mechanism. Reviewer Ka4a acknowledged our physically motivated decomposition framework and efficient nine-stage unfolded solver design.

(b) Reviewer DTCy acknowledges that our CIDNet **"revolutionizes dual-camera CASSI reconstruction"** and recognizes the good performance achieved by our method.

However, Reviewer DTCy raises concerns regarding the theoretical justification for PAN image approximation and contribution assessment. We provided theoretical analysis proving that PAN images can effectively serve as intensity components, **which gained the reviewer's recognition**. In subsequent discussions, the reviewer raised concerns about low-light scenarios, we responded with both theoretical justification and ablation studies demonstrating our method's robustness, and await the reviewer's further feedback.

(c) Reviewer 3wmZ recognizes our work as a "novel, interpretable, physically-based method" and acknowledges our superior performance in chromaticity reconstruction compared to existing methods.

After systematically addressing the reviewer's technical and methodological concerns with additional experiments, they stated **"They resolve my questions, so I will move my rating up"** and increased their score.

---

### Decision · Program_Chairs · 2025-09-17

**Decision:**

Accept (poster)

**Comment:**

The submission introduces a new framework for dual-camera compressive hyperspectral imaging. All reviewers recommended acceptance of the submission, and found the rebuttal to have mostly addressed their concerns. The area chair agrees with the reviewers' consensus, and strongly recommends the authors to incorporate the rebuttal and discussions in the camera ready.